# Fidel-TS: A High-Fidelity Benchmark for Multimodal Time Series Forecasting

## Abstract

The evaluation of time series forecasting models is hindered by a critical lack of high-quality benchmarks, leading to a potential illusion of progress. Existing datasets suffer from issues ranging from pre-training data contamination in the age of LLMs to the temporal and description leakage prevalent in early multimodal designs. To address this, we formalize the core principles of **high-fidelity benchmarking**, focusing on data sourcing integrity, leak-free and causally sound design, and structural clarity. We introduce **Fidel-TS**, a new large-scale benchmark built from the ground up on these principles by sourcing data from live APIs. Our extensive experiments validate this approach by exposing the critical biases and design limitations of prior benchmarks. Furthermore, we conclusively demonstrate that the causal relevance of textual information is the key factor in unlocking genuine performance gains in multimodal forecasting, which lead to our future works.

## 1 Introduction

The field of time series forecasting has seen relentless progress, with a proliferation of methods from classic statistical models (Box et al., 2015) and machine learning models (Chen & Guestrin, 2016) to sophisticated deep learning architectures (Hochreiter & Schmidhuber, 1997; Vaswani et al., 2017) and foundation models (Garza et al., 2023; Rasul et al., 2023). As models demonstrate increasingly strong capabilities, the field of time-series forecasting calls for evaluations that are broader, fairer, and more aligned with real-world scenarios. However, we argue that **the benchmarks currently in use have not kept pace with the rapid advancements in model development**.

This weak foundation is rooted in legacy issues and compounded by new, misguided approaches. Classic unimodal benchmarks (Zhou et al., 2021; Wu et al., 2021; 2023) are often small in scale, outdated, and feature a vague composition of variables, making them scientifically unsound for evaluating modern models. More critically, the widespread availability of these datasets has created a significant, often unacknowledged, risk of pre-training contamination for large language models (LLMs) (Magar & Schwartz, 2022; Jiang et al., 2024), which may lead to the hallucination that LLMs excel in time series forecasting. As the community has pivoted to multimodal forecasting (Williams et al., 2024; Wang et al., 2024a; Xu et al., 2024a), first-generation benchmarks (Liu et al., 2024a) have perpetuated this contamination risk while introducing equally severe flaws. The common practice of retrospectively retrieving textual data introduces severe data leakage, including temporal leakage (accessing information published after the event to be predicted) and description leakage (retrieving text that explicitly states the ground-truth value). Furthermore, relying on low-frequency data (e.g., monthly) makes these datasets too small to effectively train domain-specific numerical models and prevents them from providing sufficiently complex testing scenarios.

To rectify these shortcomings and establish a more credible evaluation standard, we reconsider the essential characteristics of data encountered in real-world forecasting scenarios. Based on this review, we formalize core principles of **high-fidelity benchmarking**: (1) **Data Sourcing Integrity**, which mandates the use of real-time, authentication-protected API data streams to not only ensure recency and data sufficiency but also mitigate pre-training contamination; (2) **Leak-free and Causally Sound Design**, which incorporates only verifiably exogenous textual information, such as weather forecasts and scheduled maintenance, to prevent temporal and description leakage; (3) **Structural Clarity**, which implements a clear demarcation between forecasting "Subjects" and data "Channels" to enable rigorously evaluating model generalization. Guided by these paradigms, we introduce **Fidel-TS**, a

new benchmark designed from the ground up to embody these principles. Each dataset contains millions of fresh, high-frequency and leak-free data points from real-time APIs, features textual information covering every timestamp, and is built with a highly extensible architecture.

To demonstrate the value of our high-fidelity benchmark, we developed a comprehensive framework compatible with models of different modalities, architectures, and input. Our experiments reveal substantial differences in the performance of many state-of-the-art models on Fidel-TS compared to prior benchmarks, and further uncover the poor performance of LLMs on forecasting tasks on Fidel-TS, indicating the evaluation bias and design flaws of prior benchmarks. We also investigate how different textual inputs affect multimodal forecasting. Our results show that performance gains hinge on the causal relevance of the text, underscoring causal alignment as an important factor for designing robust datasets and effective models. In summary, our contributions are threefold:

- A formalization of the principles for **high-fidelity** time series benchmarking.
- The release of **Fidel-TS**, a new large-scale benchmark built on these principles.
- A **thorough experimental analysis** using our proposed framework, offering new insights into the capabilities of modern forecasting models.

## 2 RELATED WORKS

### 2.1 CLASSIC UNIMODAL BENCHMARKS

The evaluation of time series forecasting models has historically relied on a narrow set of classic unimodal benchmarks, typically including ETT(ETTh1, ETTh2, ETTm1, ETTm2), ECL (Zhou et al., 2021); Electricity, Weather, Traffic, ILI (Wu et al., 2021); M4 (Wu et al., 2023), etc. The scientific utility of classic benchmarks is compromised by three core issues rooted in their age and static nature:

First, they are **small in scale**, with most datasets containing only on the order of tens of thousands of data points. This limited data volume is often insufficient for robust model evaluation, meaning that observed performance gains may be statistically insignificant or biased.

Second, they are **outdated** and have been widely accessible online for years. For instance, the ETT dataset concluded in 2018 and the Electricity dataset concluded in 2019. This long-term availability poses a significant risk of pre-training contamination, particularly for LLMs, casting doubt on the fairness of evaluations on modern LLM-based models.

Third, their **variable composition is vague**. There is no clear demarcation between datasets containing multiple variables from a single system and those featuring a single variable type across multiple independent systems. For example, the Weather dataset comprises various meteorological variables for atmosphere, whereas the Electricity and Traffic datasets treat sensors from different locations as distinct variables, despite them all measuring the same physical quantity.

### 2.2 NOVEL MULTIMODAL BENCHMARKS

While the shift to multimodality was conceptually correct, transitioning from unimodal to multimodal benchmarks is a non-trivial challenge. Classic unimodal datasets, due to their age and lack of rich contextual metadata, are difficult to extend. The initial multimodal time series forecasting benchmark, TimeMMD (Liu et al., 2024a), addressed this problem but highlighted the challenges of maintaining real world fidelity, demonstrated several drawbacks:

First, the majority of its datasets are plagued by **low sampling frequencies**. Six datasets are sampled monthly, including Agriculture, Climate, Economy, Security, Social Good, and Traffic, while Energy and Health are sampled weekly. This results in a severe scarcity of data points, with many series containing only a few hundred observations despite spanning decades. These simple samples cannot support the evaluation of complex scenarios or the training of numerical domain-specific models.

Second, the reliance on an Agent pipeline to retrieve web-based text introduces multiple, critical forms of data leakage. This method risks **pre-training contamination**, as static web content is likely already part of the training corpora for LLMs. It also leads to **temporal leakage**, where retrieved documents like future financial reports contain information unavailable at prediction time,

and **description leakage**, where text explicitly describes future trends, effectively revealing the ground truth. Furthermore, this approach fails to ensure strict temporal alignment between the textual data and its corresponding time series timestamp, increasing the risk of temporal leakage issue.

Third, it perpetuates the **ambiguous variable structure** of classic unimodal benchmarks. It also fails to distinguish between datasets comprising multiple, distinct variables from a single system versus those with a single variable type from multiple systems. Besides, in its multivariate datasets, it arbitrarily designates one channel as the "target," a design choice that oversimplifies the problem and ignores the equal importance, or potential interdependencies among variables.

Taken together, these pervasive issues of data scarcity, data leakage, and ambiguous structure across existing benchmarks result in a significant loss of real-world fidelity. This foundational weakness affects the reliable evaluation of all models and hinders credible progress. To rectify this, we argue for a new evaluation paradigm founded on a clear set of principles, which we introduce next.

## 3 FIDEL-TS: A BENCHMARK BUILT ON HIGH-FIDELITY PRINCIPLES

### 3.1 CORE PRINCIPLES OF HIGH-FIDELITY DESIGN

Our approach to achieving real-world fidelity is built on a set of interconnected principles, starting with how data is sourced and managed.

The foundation of our benchmark is the exclusive use of continuously updated data streams from **authentication-protected APIs**, eschewing the static, web-scraped files common in other benchmarks. This single architectural choice directly addresses several requirements for realism. First, it provides a stream of large quantity, fresh, high-frequency data, ensuring the benchmark remains current and can be used to evaluate models on fine-grained, sub-hourly tasks. This API-driven approach also guarantees precise temporal alignment across all modalities, a critical factor for accurate forecasting. Second, this methodology provides a multi-layered defense against information leakage. We mitigate the risk of pre-training contamination through two primary mechanisms. First, by sourcing data directly from APIs rather than from static, widely-indexed public files, we reduce the likelihood that our benchmark is part of common training corpora. More critically, the recency of our data streams, which are continuously updated and extend well beyond the typical knowledge cutoffs, serves as a robust safeguard against direct dataset memorization.

With a foundation of fresh, well-aligned data, our next principle is to populate it with leak-free, causally sound and context-rich textual information. We deliberately focus on **weather reports** (sourced from weather APIs) as the primary textual modality for the majority of our datasets. This choice is driven by three critical advantages regarding data integrity and causality. First, unlike economic reports or news updates, weather forecasts can be known in advance. This verifiable availability prevents temporal leakage. Second, weather data effectively circumvents description leakage. Whereas some textual inputs might explicitly describe the target variable (e.g., "power usage will rise next week"), weather reports strictly describe environmental conditions without directly stating the outcome of the variable itself. Finally, weather is causally sound and verifiable: based on real-world common sense, weather is a universal driver that significantly influences system dynamics, yet it remains an exogenous factor independent of and unaffected by the systems it influences.

To accurately reflect the structure of real-world data collection, we also formalize the **distinction between subjects and channels**. Our data sources, though diverse, share a common organizational structure: data originates from multiple subjects (e.g., regions, devices, or sensors), with each Subject comprising an identical set of channels (i.e., the observed variables). For example, in an electricity consumption dataset, each household is a Subject, while the measurements for 'usage (kWh)' and 'voltage (V)' are the Channels. A channel, therefore, refers to a distinct observed variable of a system, while a subject refers to a specific instance of that system. This clear distinction is pivotal for evaluating the generalization ability of models, as it better reflects real-world scenarios such as deploying a trained model to a newly added sensor within the same system.

Finally, the entire framework is designed for **extensibility**. By providing detailed metadata and documentation for all API sources, we enable researchers to easily integrate additional data streams—such as new economic indicators or social media trends. This mirrors the real-world requirement for

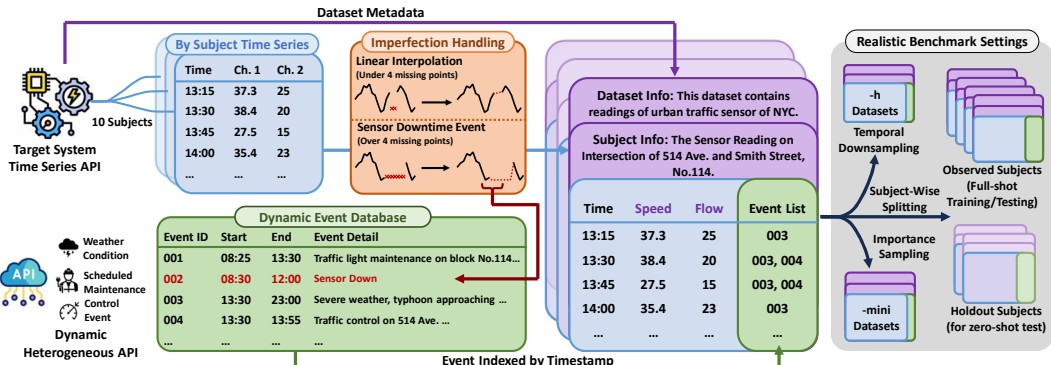

Figure 1: **The Construction Pipeline of Fidel-TS**. The workflow integrates raw data from the Target System Time Series API and the Dynamic Heterogeneous API. The Imperfection Handling step interpolates short data gaps while converting long downtime into time-aligned 'Sensor Downtime'. These downtime events and heterogeneous textual data are stored in the Dynamic Event Database together, strictly following the start and end times. All these events and time series, combined with static dataset metadata, forms a unified dataset that is systematically organized into Realistic Benchmark Settings: downsampled (`-h`), sampled via importance (`-mini`), observed (`-obs`, for in-domain evaluation), and hidden (`-hid`, for generalization evaluation).

forecasting systems to adapt and scale as new, relevant information becomes available, ensuring the long-term relevance and utility of the benchmark.

## 3.2 Data Curation Pipeline: From Raw Streams to a High-Fidelity Benchmark

Translating raw, real-time data streams into a structured benchmark requires a pipeline that preserves the nuances of real-world data. The following steps detail our process, with a full summary of the complete pipeline in Figure 1 and the resulting datasets in Table 1. More detailed background information of these datasets are provided in Appendix A.1.

**Handling Real-World Data Imperfections.** Embodying our principle of high fidelity, a core feature of the benchmark data is that it directly reflects the challenges of real-world instrumentation. Sourced from physical sensors rather than statistically processed tables, the raw time series data naturally contains imperfections like environmental noise and missing values. This inherent instability is a feature, not a flaw, representing the unfiltered reality of live data streams. Our first step involves filtering obvious outliers and aligning timestamps to a consistent sampling rate. To handle missing values realistically, we distinguish between short and long gaps. Short-duration gaps are filled using linear interpolation, reflecting minor data transmission issues. However, longer gaps are treated as significant real-world events; we mark the numerical values as zero and generate a corresponding textual description, such as 'sensor downtime', which is aligned with the relevant timestamps. This transforms a data quality issue into a valuable contextual feature for the models to learn from.

**Enriching Data with Static and Dynamic Textual Context.** Real-world forecasting requires both foundational knowledge and real-time updates. To provide this, we integrate two distinct types of textual information. First, we establish the static context by sourcing background knowledge from the official API metadata: a general overview for the entire dataset, specific descriptions for each Subject, and detailed explanations for each Channel. Second, we incorporate dynamic, time-aligned information that reflects the evolving state of the system and its environment. This dynamic information, along with detected sensor downtime events in Imperfection Handling step, is consolidated into the Dynamic Event Database, which features storage indexed by start and end times, ensuring that the textual context is strictly aligned with the timestamps of the numerical time series. For all datasets, we provide weather reports for their respective locations. For relevant datasets, we also support the use of control information extracted from system logs or scheduled events. See Appendix A.2 for examples and analysis of our preprocessed time series and textual data.

Table 1: Overview of Dataset Statistics in Fidel-TS

| Dataset | Time Span | Samp. Freq. | No. Subject | No. Cha. (per Sub.) | Textual Data Type | Textual Data Frequency | Total Data Points |
|---|---|---|---|---|---|---|---|
| **Canada Calgary Photovoltaics (CCP)** | 2018-01-01 ∼ 2025-02-28 | 1h | 9 | 1 | Weather | 6h, 1day | 564,881 |
| **Germany Renewable Energy Grid (GREG)** | 2015-01-01 ∼ 2025-02-25 | 15min | 8 | 1 | Weather | 6h, 1day | 2,848,472 |
| **Jena Atmospheric Physics (JAP)** | 2014-01-01 ∼ 2025-06-30 | 10min | 1 | 21 | Weather | 6h, 1day | 604,311 |
| **NYC Traffic Speed (NYTS)** | 2018-01-01 ∼ 2025-03-01 | 5min | 97 | 1 | Weather | 6h, 1day | 67,517,110 |
| **California ISO (CAISO)** | 2018-04-10 ∼ 2025-02-28 | 5min | 1 | 20 | Weather | 6h, 1day | 724,896 |
| **UCSD BEAR Room (BEAR)** | 2020-06-01 ∼ 2020-10-20 | 5min | 80 | 3 | Weather Control Event | 6h, 1day 5min | 3,156,240 |

**Designing Evaluation Scenarios.** To test model performance under diverse real-world conditions, we create several dataset variations. First, to ensure a fair comparison across datasets with different native sampling frequencies, we provide an option to downsample high-frequency series to a uniform hourly level (denoted by the `-h` suffix). Second, to address the unique computational challenges of evaluating LLMs, we curate smaller subsets (denoted by the `-mini` suffix) constructed via importance sampling. By selecting samples from the full test set where unimodal models outperform multimodal ones and vice versa, we ensure both computational efficiency and fairness. More details are provided in Appendix A.3.

**Subsetting for Generalization and Transferability.** A crucial aspect of real-world utility is a model's ability to generalize to new scenarios it has not been explicitly trained on. Our Subject and Channel definitions naturally support this evaluation. For datasets with a large number of Subjects, we create distinct partitions to simulate a zero-shot learning environment. Subsets intended for standard training and evaluation are marked with the `-obs` (observed) suffix, while those held back for testing generalization on unseen subjects are marked with the `-hid` (hidden) suffix, providing a robust test of a model's transfer learning capabilities. More details are provided in Appendix A.4.

To summarize the advancement of our benchmark, we also provide a comparison between Fidel-TS and previous benchmarks in Appendix A.5.

## 4 EXPERIMENTS

### 4.1 EVALUATION FRAMEWORK

The high-fidelity nature of our Fidel-TS benchmark establishes a robust foundation for evaluation. By supplying rich, leakage-free data, it enables not only a critical re-assessment of traditional unimodal models but also a sound basis for evaluating modern multimodal models and LLMs. However, the forecasting model landscape has become increasingly heterogeneous, with vast differences in model scale, architectural backbones, and input-output definitions. This diversity highlights a pressing need for a universal framework that enables fair and convenient comparisons.

Existing toolkits (such as LSTF-Linear (Zeng et al., 2023), TSLib (Wang et al., 2024b), MM-TSFLib (Liu et al., 2024a), TSFM-Bench (Li et al., 2025)) often tailored to specific model categories and lack the flexibility to accommodate this wide spectrum of models under a single, unified configuration. To bridge the gap, we developed our **Universal Cross-modal Evaluation Framework** designed to connect our high-fidelity data with the diverse model ecosystem. The comparison in Table 2 illustrates the comprehensiveness of our benchmark across training paradigms, model types, modality support, and evaluation strategies. Our framework provides an integrated testing environment for diverse models and facilitates unified experimental configuration. More implementation details of our framework are provided in Appendix B.

Table 2: Comparison of time series forecasting frameworks across four key dimensions

| Benchmarks | Training Paradigms | | Model Type | | | Modality Support | | Evaluation Strategy | |
|---|---|---|---|---|---|---|---|---|---|
| | Domain-specific | Cross-domain | Numerical | LLM-based | LLM API | Uni-modal | Multi-modal | Full-shot | Zero-shot |
| **LSTF-Linear** | ✓ | | ✓ | | | ✓ | | ✓ | |
| **TSLib** | ✓ | | ✓ | | | ✓ | | ✓ | |
| **TSFM-Bench** | ✓ | ✓ | ✓ | ✓ | | ✓ | | ✓ | ✓ |
| **MM-TSFLib** | ✓ | | ✓ | ✓ | ✓ | ✓ | ✓ | ✓ | |
| **Ours** | ✓ | ✓ | ✓ | ✓ | ✓ | ✓ | ✓ | ✓ | ✓ |

## 4.2 EXPERIMENTAL SETUP

**Model Selection**    Using our new-proposed framework, we conduct a comprehensive evaluation across representative models, which we broadly categorize into:

(1) **Unimodal Forecasting Models**, including MLP-based Dlinear (Zeng et al., 2023), FITS (Xu et al., 2024b), transformer-based PatchTST (Nie et al., 2023), iTransformer (Liu et al., 2024b), LLM-based GPT4TS (Zhou et al., 2023), and advanced foundation models Chronos (Ansari et al., 2024), TimeMoE (Shi et al., 2025) and Sundial (Liu et al., 2025), which pre-trained on vast numerical time series corpora.

(2) **Multimodal Forecasting Models**, including LLM-based GPT4MTS (Jia et al., 2024) and transformer-based FIATS (Xu et al., 2024a). They have different input requirements due to their distinct modeling approaches.

(3) **LLMs**, including general-purpose LLMs Qwen-2.5-14B-Instruct (Yang et al., 2024), Qwen-2.5-14B-Instruct-1M (Yang et al., 2025b), Qwen-3-14B (Yang et al., 2025a), Deepseek-R1 (DeepSeek-AI et al., 2025), and expert LLM Chattime (Wang et al., 2025) which fine-tuned based on Llama-2-7b (Touvron et al., 2023). To accommodate LLMs for forecasting, we designed prompts for both unimodal and multimodal forecasting tasks.

See Appendix C for more details of models.

**Evaluation Metrics**    We adopt the **Mean Square Error (MSE)** as the main evaluation metric:

$$\text{MSE} = \frac{1}{n} \sum_{i=1}^{n} (y_i - \hat{y}_i)^2 \tag{1}$$

We added a **pass rate** to evaluate the LLMs, defined identically to the pass@k metric used for tasks like code generation or math problem-solving. This addresses cases where the LLM occasionally disregard prompt constraints, causing format errors (e.g., incorrect prediction length or variable numbers) and task failure. Using k=3, we report the pass rate within three attempts across the test set.

**Realistic Settings**    To simulate diverse real-world time scales, our design aligns forecasting horizons with intuitive and operational periods. For instance, for datasets with an hourly frequency, we employ historical horizon of 360 (indicates half a month) and prediction lengths of {24, 168, 336, 720} which correspond to {1 day, 1 week, 2 weeks, and 1 month}. Our framework not only supports classic ratio-based splitting (e.g., 7:1:2) (Zhou et al., 2021; Wu et al., 2021), but also enables splitting by timestamps (e.g. Jan. 1st, 2021, Jan. 1st, 2022). This approach makes the evaluation setup more transparent and controllable, providing further safeguard against potential leakage issue. See Appendix D for more details of settings.

## 4.3 BENCHMARKING UNIMODAL FORECASTING

The evaluation results of all unimodal forecasting models on Fidel-TS are presented in Table 3. The results show that: (1) Although most of the domain-specific models have previously claimed state-of-the-art status on classic unimodal benchmarks, the performance of them is highly dataset-dependent

Table 3: Performance of unimodal forecasting models on the full test set of each dataset. All metrics are Mean Squared Error (**MSE**). The best results are highlighted in **red** and the second-best in blue. Domain-specific models are trained on the training set of each dataset. Foundation models are evaluated in zero-shot forecasting without fine-tuning on each training set.

| Dataset | Pred. Len. | Domain-Specific | | | | | Foundation Models | | |
|---|---|---|---|---|---|---|---|---|---|
| | | PatchTST | iTrans. | Dlinear | FITS | GPT4TS | Chronos | Time-MoE | Sundial |
| CCP-h | 24 | 0.159 | **0.147** | 0.158 | 0.161 | 0.149 | 0.152 | 0.168 | 0.167 |
| | 168 | 0.205 | **0.186** | 0.205 | 0.205 | 0.193 | 0.205 | 0.218 | 0.227 |
| | 336 | 0.224 | **0.197** | 0.221 | 0.223 | 0.206 | 0.229 | 0.246 | 0.246 |
| | 720 | 0.258 | **0.218** | 0.253 | 0.254 | 0.236 | 0.288 | 0.362 | 0.275 |
| GREG-h | 24 | 0.094 | 0.090 | 0.100 | 0.099 | 0.088 | **0.084** | 0.095 | 0.090 |
| | 168 | **0.130** | 0.132 | 0.151 | 0.131 | 0.131 | 0.137 | 0.144 | 0.142 |
| | 336 | 0.139 | **0.138** | 0.172 | 0.141 | 0.139 | 0.152 | 0.164 | 0.151 |
| | 720 | 0.163 | **0.155** | 0.199 | 0.167 | 0.157 | 0.188 | 0.260 | 0.172 |
| JAP-h | 24 | 0.362 | **0.356** | 0.376 | 0.382 | 0.358 | 0.360 | 0.359 | 0.361 |
| | 168 | 0.541 | 0.546 | 0.544 | 0.553 | **0.540** | 0.670 | 0.565 | 0.592 |
| | 336 | 0.591 | 0.593 | **0.586** | 0.599 | 0.591 | 0.723 | 0.630 | 0.634 |
| | 720 | 0.663 | 0.663 | **0.647** | 0.674 | 0.668 | 0.820 | 0.735 | 0.706 |
| CAISO-h | 24 | 0.183 | 0.186 | 0.190 | 0.193 | 0.183 | 0.183 | **0.177** | 0.181 |
| | 168 | **0.360** | 0.386 | 0.392 | 0.368 | 0.366 | 0.389 | 0.377 | 0.382 |
| | 336 | **0.429** | 0.452 | 0.486 | 0.437 | 0.444 | 0.473 | 0.464 | 0.460 |
| | 720 | **0.536** | 0.555 | 0.655 | 0.550 | 0.542 | 0.608 | 0.610 | 0.577 |
| NYTS-h-obs | 24 | 0.601 | **0.581** | 0.638 | 0.644 | 0.582 | 0.613 | 0.680 | 0.649 |
| | 168 | **1.026** | 1.041 | 1.098 | 1.125 | 1.065 | 1.177 | 1.210 | 1.275 |
| | 336 | **1.176** | 1.195 | 1.262 | 1.306 | 1.194 | 1.512 | 1.421 | 1.497 |
| | 720 | **1.275** | 1.295 | 1.378 | 1.457 | 1.329 | 1.799 | 1.592 | 1.646 |
| BEAR-obs | 12 | 0.121 | **0.105** | 0.128 | 0.169 | 0.113 | 0.151 | 0.147 | 0.141 |
| | 144 | 0.483 | **0.416** | 0.506 | 0.582 | 0.474 | 1.056 | 0.765 | 0.787 |
| | 288 | 0.680 | **0.619** | 0.650 | 0.767 | 0.652 | 1.312 | 1.012 | 1.043 |
| | 576 | 0.874 | **0.802** | 0.837 | 1.039 | 0.937 | 1.672 | 1.299 | 1.353 |

on our benchmark, with no single model consistently emerging as a front-runner. (2) while large-scale foundation models claim to "fit all time series" and can outperform domain-specific models in zero-shot forecasting, their performance is merely comparable to specialized models on short forecast horizons and degrades significantly over longer ones. These suggest that the improvements observed on classic unimodal benchmarks are likely biased and do not generalize, questioning their validity in assessing the true capability of the model. See more results and analysis in Appendix E.1.

Table 4: Performance of LLMs (for unimodal forecasting) on `-mini` test set of each dataset. All metrics are **MSE**; The best results are highlighted in **red** and the second-best in blue. "–" indicates the context for the forecasting task exceeded the LLM's input limit, or the LLM failed on all samples due to incorrect output format.

| Dataset | Pred. Len. | LLMs (Unimodal forecasting) | | | | | Domain-Specific | | | | | Foundation Models | | |
|---|---|---|---|---|---|---|---|---|---|---|---|---|---|---|
| | | Qwen2.5-14B Instruct | Qwen2.5-14B Instruct-1M | Qwen3-14B | Chattime | Deepseek-R1 | Patch. | iTrans. | Dlin. | FITS | GPT4TS | Chronos | Time-MoE | Sundial |
| CCP-mini | 24 | 0.311 | 0.289 | 0.235 | 0.263 | 0.182 | 0.140 | **0.127** | 0.138 | 0.141 | 0.129 | 0.144 | 0.146 | 0.131 |
| | 168 | 0.865 | 0.374 | 0.298 | 0.457 | 0.267 | 0.192 | **0.169** | 0.197 | 0.197 | 0.179 | 0.203 | 0.219 | 0.192 |
| GREG-mini | 24 | 0.229 | 0.213 | 0.153 | 0.223 | 0.113 | 0.073 | **0.070** | 0.089 | 0.086 | 0.090 | 0.093 | 0.089 | 0.085 |
| | 168 | 0.497 | 0.279 | 0.263 | 0.486 | 0.213 | **0.135** | 0.135 | 0.150 | 0.137 | 0.143 | 0.170 | 0.162 | 0.154 |
| JAP-mini | 24 | – | 1.825 | – | 0.483 | – | 0.286 | 0.285 | 0.334 | 0.334 | 0.278 | **0.277** | 0.278 | 0.286 |
| | 168 | – | – | – | 0.580 | – | 0.418 | **0.411** | 0.419 | 0.429 | 0.421 | 0.443 | 0.449 | 0.448 |
| NYTS-mini | 24 | 1.884 | 1.893 | 2.236 | 0.998 | 1.071 | 0.826 | **0.764** | 0.864 | 0.871 | 0.774 | 0.880 | 0.923 | 0.777 |
| | 168 | 3.599 | 2.378 | 2.546 | 2.312 | 2.154 | **1.538** | 1.599 | 1.560 | 1.615 | 1.749 | 1.876 | 1.675 | 1.797 |
| CAISO-mini | 24 | – | 0.501 | – | 0.216 | – | 0.136 | 0.140 | 0.143 | 0.145 | 0.137 | 0.146 | 0.132 | **0.130** |
| | 168 | – | – | – | 0.453 | – | **0.276** | 0.297 | 0.324 | 0.285 | 0.278 | 0.297 | 0.298 | 0.282 |
| BEAR-mini | 12 | 0.944 | 0.522 | 4.279 | 0.145 | 0.299 | 0.103 | **0.080** | 0.111 | 0.144 | 0.091 | 0.126 | 0.118 | 0.116 |
| | 144 | 2.205 | 7.425 | 1.486 | 1.077 | 2.249 | 0.484 | **0.353** | 0.505 | 0.593 | 0.432 | 0.836 | 0.831 | 1.027 |

The evaluation results of LLMs for unimodal forecasting (combined with the test results of unimodal forecasting models on `-mini` datasets) are summarized in Table 4. The results show that despite prior claims of performance enhancement through prompting (Gruver et al., 2023), LLMs consistently underperform specialized unimodal models in all tested scenarios on our benchmark. This suggests that when provided solely with historical time series as input, LLMs struggle to effectively capture

temporal patterns and extrapolate numerical values for forecasting. The improvements observed in previous studies are likely artifacts of flawed evaluation, stemming from either pre-training contamination on outdated benchmarks, which allows LLMs to memorize answers, or overly short prediction horizons that render the task trivially simple.

## 4.4 GENERALIZATION TEST

Classic benchmarks often employ false transfer strategies (Jin et al., 2024), forcing models to transfer between datasets with entirely different domain (e.g., ETT → Weather), or between subsets of the same domain with mismatched sampling frequencies (e.g., ETTh1 → ETTm1). In contrast, our distinction between subjects and channels enables a more realistic transfer setup. We train models exclusively on the -obs subsets of NYTS and BEAR and evaluate them directly on the held-out -hid subsets, simulating a more practical scenario.

Table 5: Generalization test of domain-specific unimodal forecasting models. All metrics are **MSE**. The best results are highlighted in **red** and the second-best in blue.

| Dataset | Pred. Len. | PatchTST | iTrans. | Dlinear | FITS | GPT4TS |
|---|---|---|---|---|---|---|
| NYTS-h-obs → NYTS-h-hid | 24 | 0.611 | 0.606 | 0.650 | 0.655 | **0.603** |
| | 168 | **1.123** | 1.142 | 1.187 | 1.228 | 1.162 |
| | 336 | **1.310** | 1.329 | 1.374 | 1.440 | 1.325 |
| | 720 | **1.424** | 1.445 | 1.501 | 1.602 | 1.513 |
| BEAR-obs → BEAR-hid | 12 | 0.100 | **0.059** | 0.105 | 0.139 | 0.094 |
| | 144 | 0.424 | **0.382** | 0.456 | 0.507 | 0.426 |
| | 288 | 0.588 | **0.549** | 0.595 | 0.659 | 0.579 |
| | 576 | 0.762 | **0.717** | 0.801 | 0.906 | 0.876 |

As presented in Table 5, the performance of the models is inconsistent, mirroring the results in Table 3. In our generalization test which is designed to be more meaningful in real-world scenarios, no single model demonstrates superior transfer capabilities. This suggests that previous transfer strategies may also introduce bias, producing an illusion of strong transferability in unrealistic tasks, yet this effect does not hold in practical, real-world scenarios.

## 4.5 BENCHMARKING MULTIMODAL FORECASTING

Table 6: Performance of multimodal forecasting models and LLMs (for multimodal forecasting) on -mini test set of each dataset. All metrics are **MSE**; Best results are in **red**, second-best in blue. "–" indicates the context for the forecasting task exceeded the LLM's input limit, or the LLM failed on all samples due to incorrect output format. We use control events for the BEAR dataset.

| Dataset | Pred. Len. | LLMs (Multimodal forecasting) | | | | | Domain-Specific | |
|---|---|---|---|---|---|---|---|---|
| | | Qwen2.5 -14B Instruct | Qwen2.5 -14B Instruct -1M | Qwen3 -14B | Chattime | Deepseek -R1 | GPT4MTS | FIATS |
| CCP-mini | 24 | 0.315 | 0.303 | 0.222 | 0.313 | 0.137 | 0.146 | **0.093** |
| | 168 | 0.570 | 0.555 | 0.336 | – | 0.216 | 0.199 | **0.142** |
| GREG-mini | 24 | 0.267 | 0.274 | 0.210 | 0.677 | 0.130 | 0.092 | **0.047** |
| | 168 | 0.405 | 0.421 | 0.282 | – | 0.188 | 0.156 | **0.074** |
| JAP-mini | 24 | – | 1.457 | – | 0.624 | – | 0.275 | **0.234** |
| | 168 | – | – | – | – | – | 0.416 | **0.304** |
| CAISO-mini | 24 | – | 0.421 | – | 0.874 | – | **0.138** | 0.170 |
| | 168 | – | – | – | – | – | 0.303 | 0.364 |
| NYTS-mini | 24 | 2.537 | 2.562 | 3.169 | 2.455 | 1.013 | 0.838 | **0.387** |
| | 168 | 3.184 | 3.061 | 2.911 | – | 1.138 | 1.643 | **0.700** |
| BEAR-mini | 12 | – | 1.577 | – | 1.209 | – | 0.093 | **0.066** |
| | 144 | – | – | – | – | – | 0.540 | **0.211** |

The test results of multimodal forecasting models and LLMs (for multimodal forecasting) on Fidel-TS are shown in Table 6. The results show that less capable models like the Qwen series and Chattime struggle to benefit from textual information; in some cases, their multimodal performance is even worse than their unimodal, series-only counterpart (in Table 4). Although Deepseek-R1, the most capable in reasoning, performs well in some cases, it still falls short of smaller, specialized multimodal forecasting models in most cases. This result stands in contrast to previously reported successes of LLMs in multimodal forecasting (Wang et al., 2024a; Williams et al., 2024). On one hand, prior benchmarks were often susceptible to temporal and description leakage, allowing LLMs to effectively

"cheat" by extracting ground truth from text rather than genuinely learning cross-modal relationships. On our leak-free benchmark, this shortcut is eliminated, thereby exposing their weakness in this core task. On the other hand, data scarcity in previous benchmarks led to simplistic tasks with sparse text and short prediction horizons. While LLMs performed well in these easy cases, our benchmark's richer textual data and more practical, longer forecasting windows reveal a performance degradation, highlighting their struggles with complex, long-context scenarios.

The complete test results for the domain-specific multimodal forecasting models are summarized in Appendix E.2, indicating the improvement in forecasting performance of multimodal FIATS model compared to unimodal models. Appendix E.3 provides sample reasoning process of LLMs, demonstrate the differences in the reasoning process of LLMs in unimodal and multimodal contexts.

## 5 DISCUSSION

### 5.1 EFFICIENCY ANALYSIS

**LLM Reliability**  To evaluate the reliability of LLMs in long-term time series forecasting, we report their pass rate on the test set as defined in our metrics section. As shown in Table 7, failures are already evident in unimodal settings, particularly in datasets with a large number of channels where lower pass rates are observed. This instability is further exacerbated in multimodal tasks. Unlike previous evaluations that utilized sparse text and short prediction horizons, our benchmark incorporates extensive high-frequency textual data. This substantial increase in input length triggers context explosion, resulting in significantly higher failure rates under these more demanding conditions.

Table 7: Comparison of **Pass Rates (pass@3)** between Unimodal (Uni.) and Multimodal (Multi.) forecasting for LLMs.

| Dataset | Pred. Len. | Qwen2.5 -14B-Inst. 128K Context | | Qwen2.5 -14B-Inst.-1M 1M Context | | Qwen3 -14B 128K Context | | Chattime 4K Context | | Deepseek -R1 128K Context | |
|---|---|---|---|---|---|---|---|---|---|---|---|
| | | Uni. | Multi. | Uni. | Multi. | Uni. | Multi. | Uni. | Multi. | Uni. | Multi. |
| CCP-mini | 24 | 96.3% | 96.8% | 96.9% | 97.1% | 98.8% | 95.6% | 100% | 100% | 99.9% | 99.8% |
| | 168 | 75.0% | 71.4% | 73.6% | 81.5% | 90.6% | 77.2% | 100% | 0.0% | 97.5% | 94.9% |
| GREG-mini | 24 | 96.9% | 98.1% | 97.6% | 99.4% | 96.7% | 94.3% | 100% | 99.2% | 99.8% | 99.8% |
| | 168 | 74.4% | 91.6% | 73.2% | 86.4% | 88.0% | 77.2% | 100% | 0.0% | 93.2% | 97.2% |
| JAP-mini | 24 | 0.0% | 0.0% | 29.1% | 47.4% | 0.0% | 0.0% | 100% | 100% | 0.0% | 0.0% |
| | 168 | 0.0% | 0.0% | 0.0% | 0.0% | 0.0% | 0.0% | 100% | 0.0% | 0.0% | 0.0% |
| NYTS-mini | 24 | 93.2% | 95.7% | 93.7% | 97.1% | 97.7% | 96.3% | 100% | 99.9% | 99.9% | 100% |
| | 168 | 72.9% | 81.8% | 69.4% | 82.4% | 90.0% | 90.0% | 100% | 0.0% | 95.3% | 98.2% |
| CAISO-mini | 24 | 0.0% | 0.0% | 45.8% | 69.6% | 0.0% | 0.0% | 100% | 48.9% | 0.0% | 0.0% |
| | 168 | 0.0% | 0.0% | 0.0% | 0.0% | 0.0% | 0.0% | 100% | 0.0% | 0.0% | 0.0% |
| BEAR-mini | 12 | 96.7% | 0.0% | 95.2% | 63.6% | 99.8% | 0.0% | 100% | 88.3% | 99.9% | 0.0% |
| | 144 | 11.6% | 0.0% | 58.1% | 0.0% | 62.8% | 0.0% | 100% | 0.0% | 88.4% | 0.0% |

**Computation Cost**  We summarize the parameter statistics for all models in Figure 2. For domain-specific models, linear models (FITS, DLinear) remain the most efficient, followed by transformer-based models where multimodal approaches (FIATS) surpass unimodal ones (PatchTST, iTransformer) in parameter size. For the LLM-based methods (GPT4TS/GPT4MTS), while total counts are higher, the trainable parameters remain limited due to the frozen GPT-2 backbone; The multimodal GPT4MTS is slightly larger than the unimodal GPT4TS.

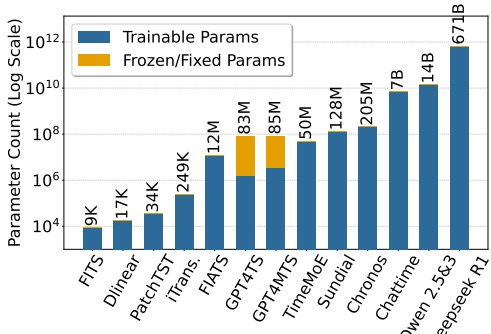

Figure 2: Comparison of model parameters

The parameter scale increases further with cross-domain pretrained Time Series Foundation Models (TimeMoE, Sundial, Chronos) and peaks with LLMs, ranging from SFT-based (Chattime) to the general LLMs (Qwen series and Deepseek-R1).

## 5.2 ABLATION ON DIFFERENT TEXTUAL INPUT

We further investigate the impact of different textual inputs on multimodal forecasting performance. The BEAR dataset offers an ideal setting for this analysis, as it uniquely contains two distinct types of textual data (external weather reports and internal control information). Comparing Table 4 and Table 6, the FIATS model demonstrates a clear advantage over unimodal baselines and achieves the best performance among multimodal forecasters. This confirms that FIATS successfully leverages textual information to enhance predictions, justifying its selection for this targeted study.

We then evaluated FIATS on the BEAR dataset using either weather data or control signals as the sole textual input, with results summarized in Table 8. The analysis reveals that control information leads to significantly better forecasting accuracy. This finding suggests that control events are the more causally relevant text modality for this system, directly contributing to the improved performance.

Table 8: Performance comparison of FIATS with different textual inputs on BEAR test set. All metrics are **MSE**.

| Textual Input | Pred. Len. | | | |
|---|---|---|---|---|
| | 12 | 144 | 288 | 576 |
| +Weather | 0.129 | 0.562 | 0.998 | 1.309 |
| +Control Event | 0.094 | 0.222 | 0.298 | 0.323 |

## 5.3 LIMITATIONS AND FUTURE WORKS

**Data Perspective**   To uphold a strict leak-free principle, we primarily utilized weather reports as the textual data; while weather is generally a causally sound factor, our ablation study suggests it may not always be the most potent driver for every system, as we discussed in the ablation study. Future work should include expanding the dataset with more causally sound textual sources while strictly adhering to our real-world fidelity principle to prevent leakage. Furthermore, Integrating these diverse inputs will require the development of robust data filtering pipelines to ensure that the context of models contains the best textual data to assist in forecasting.

**Model Perspective**   We selected representative models within the allowable range of computing resources to uncover generalized challenges within current long-term time series forecasting. Given the rapid evolution of this field, it is infeasible to cover every emerging architecture, such as specific LLM prompting methods (e.g., LLM prompting approaches LLMP(Requeima et al., 2024), LLM-Time(Gruver et al., 2023)). Future work can expand upon these baselines to include a wider array of models and specific methodologies, further demonstrating the long-term value of our benchmark.

## 6 CONCLUSION

In this work, we addressed a critical flaw in the evaluation of time series forecasting models by formalizing the principles of high-fidelity benchmarking and introducing Fidel-TS, a new large-scale benchmark built upon them. Our comprehensive experiments validate this new paradigm, leading to two core insights: (1) our high-fidelity benchmark reveals a strikingly different model performance landscape from prior work, exposing significant biases in conventional benchmarks that stemmed from unrealistic evaluation setups and data leakage. (2) we demonstrate that causal relevance is a critical determinant of success in multimodal forecasting, where the value of auxiliary data is dictated by its causal connection to the target system. By providing a robust and causally-sound evaluation standard, Fidel-TS offers a clearer path for the community to develop the next generation of truly effective forecasting models.

## ETHIC STATEMENT

This paper presents work whose goal is to advance the field of Machine Learning. There are many potential societal consequences of our work, none which we feel must be specifically highlighted here.

We comply with intellectual property agreements for all data sources. Data are properly anonymized and content generated by OpenAI API is free for general use, with no concerns regarding sensitive or illegal activity in our dataset.

All data sources utilized in this work are either under a CC0 license or from APIs with terms of service that permit academic research.

## REPRODUCIBILITY STATEMENT

The code for this work is available at https://anonymous.4open.science/r/Universal-Cross-Modal-Time-Series-Forecasting-Pipeline-7ACD. Our dataset has been made publicly available on the HuggingFace Hub. However, to comply with the double-blind review policy, we cannot provide a direct link, and the substantial size of the dataset makes a fully anonymous copy infeasible. We will include a small but representative sample of the data within our codebase.

## LLM USAGE STATEMENT

We used LLMs for grammar checking and rephrasing, without contributing to the core ideas and methodologies.

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

APPENDIX

# A BENCHMARK DETAILS

## A.1 METADATA AND DESCRIPTIONS

**Canada Calgary Photovoltaics**   1. General Information: This dataset provides hourly solar power generation data, measured in kilowatt-hours (kWh), from a collection of solar panels located in Calgary, Alberta, Canada. 2. Subjects (9): A Subject in this dataset represents an individual solar panel installed at a specific public facility in Calgary. Each subject is identified by a unique ID (e.g., 314106, 319086). The metadata provides the precise location for each panel, such as "Calgary Fire Hall Headquarters" or "Southland Leisure Centre". 3. Channels (1): Each subject has a single Channel, which is the Solar Power Generation measured in kWh. 4. Textual Data: The primary source of textual information for this dataset is weather data for Calgary, Alberta.

**Germany Renewable Energy Grid**   1. General Information: This dataset contains time-series data on the renewable energy supply from four major transmission system operators (TSOs) in Germany. It captures the generation from both solar and wind sources. 2. Subjects (8): Each Subject represents a specific energy source from one of the German power grids. For example, the subject "solar_50Hertz" represents the solar power generation from the electricity transmission system in the north and east of Germany. 3. Channels (1): Each subject has a single Channel, representing the Energy Generation for that specific source and grid (unit is typically Megawatts, MW). 4. Textual Data: Textual information consists of regional weather data for Germany, aligned to the specific regions served by each TSO, including Bayreuth, Berlin, Dortmund, Stuttgart.

**Jena Atmospheric Physics**   1. General Information: This dataset consists of comprehensive atmospheric physical indicators in Jena, Germany. It is a dataset for meteorological time-series forecasting. 2. Subjects (1): There is only one Subject, which represents the entire Jena weather station as a single system. 3. Channels (21): The Channels are the 21 distinct meteorological variables measured by the station. There are some key variables such as: Carbon dioxide concentration (measured in ppm), Actual vapor pressure (measured in mbar). 4. Textual Data: The textual information is sourced from external weather forecast in Jena, Germany.

**NYC Traffic Speed**   1. General Information: The data is collected by traffic sensors deployed across various road segments in New York City, including information about the vehicles on the roads. 2. Subjects (97): A Subject is a unique traffic sensor monitoring a specific road segment in one of the NYC boroughs (e.g., "Queens, with segment of CIP N TNB - Whitestone Expwy S Exit 14 (Linden Pl)"). Each sensor is identified by a unique ID (e.g., 204, 184). 3. Channels (1): Each subject has a single Channel, which is the Average Vehicle Speed in km/h. 4. Textual Data: The textual data is composed of weather information for New York City, covering New York, Queens and Brooklyn.

**California ISO**   1. General Information: This dataset provides a holistic view of the California electricity grid, managed by the California Independent System Operator (CAISO). 2. Subjects (1): There is only one Subject, representing the entire CAISO grid system as a single, interconnected entity. 3. Channels (20): The Channels are the 20 distinct metrics that describe the state of the grid. These include demand, power generation from various sources (Hydro, Nuclear, Biomass, etc.), and $CO_2$ emissions. 4. Textual Data: Textual information is derived from state-wide weather data for California, including San Francisco and San Diego.

**UCSD BEAR Room**   1. General Information: This dataset contains HVAC (Heating, Ventilation, and Air Conditioning) relevant data collected from 80 rooms within a single building (NAE-01) at the University of California San Diego. 2. Subjects (80): A Subject corresponds to an individual room within the building. Each room is identified by its unique room number (e.g., 104, 208). The metadata describes each room's specific location. 3. Channels (3): Each subject (room) has a consistent set of three Channels: Zone Temperature, Real Power, and Supply Flow. 4. Textual Data: This dataset uniquely features two types of textual information. The primary external source is weather data for San Diego, California. Additionally, it includes internal control event logs (e.g., 'scheduled occupation of a room').

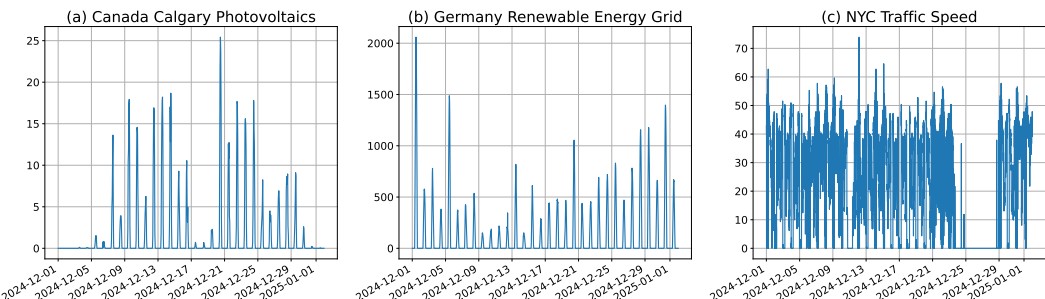

Figure 3: Visualization of the time series data in three datasets with a single channel. We select a representative time periods to show the patterns of the time series in each dataset.

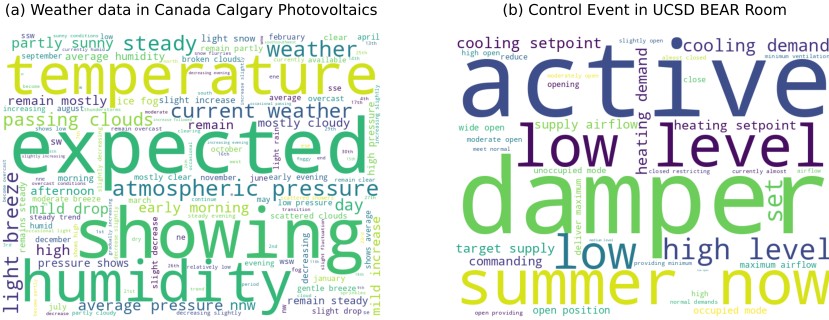

Figure 4: Visualization of the textual data. We select representative weather data and control events.

## A.2 VISUALIZATION

**Time Series Data** For illustrative purposes, we selected datasets with a single channel and plotted one of their subjects over a fixed time range. It can be observed from Figure 3 that the CCP and GREG datasets exhibit clear periodicity within the selected range. The NYTS dataset is not only noisier and exhibits more frequent, irregular fluctuations, leading to greater prediction difficulty, but also includes periods of sensor downtime, resulting in significant sudden distributional shifts. This suggests that predicting on the NYTS dataset is more challenging and will likely result in higher errors compared to the CCP and GREG datasets.

**Textual Data** We also provide word clouds generated from one year of weather data in the CCP dataset and the control events from a single room in the BEAR dataset. As illustrated in Figure 4, these visualizations clearly highlight the main content of each text source.

## A.3 IMPORTANCE SAMPLING METHOD FOR LLM EVALUATION

Evaluating LLMs on the full test set presents prohibitive computational costs due to the high frequency and large scale of the data. To address this while ensuring the evaluation remains unbiased, we constructed a representative subset (denoted as -mini) using the importance sampling strategy.

We observed that samples within the test set exhibit varying degrees of multimodal dependency: for some instances, textual information is highly correlated with the time series and significantly aids forecasting, while for others, it offers marginal gains. Simple random sampling risks distorting this distribution. To preserve the original "difficulty" distribution, we employed two lightweight models: a representative unimodal model (PatchTST, Nie et al. (2023)) and a multimodal model (FIATS, Xu et al. (2024a)). By evaluating both on the full test set, we categorized samples into two groups based on the performance gap: (1) Text-sensitive samples: Instances where the multimodal model significantly outperformed the unimodal one. (2) Text-insensitive samples: Instances where the improvement was negligible.

Finally, we applied stratified sampling with a fixed ratio to each group. Based on the data volume of each dataset, the final selected ratios are: 1% per subject for NYTS, 5% per subject for BEAR, and 10% per subject for other datasets. This ensures an appropriate data volume for the final -mini test set, while test set faithfully reflects the multimodal sensitivity distribution of the original data, providing a reliable and efficient benchmark for LLM evaluation.

### A.4 SUBSETTING METHOD FOR GENERALIZATION EVALUATION

**NYTS**  The NYTS dataset comprises 97 subjects, representing traffic speed sensors distributed across New York City. Among these, 10 are newly installed sensors with data available only since August 2021. We hold out these 10 subjects to form the `-hid` subset, while the remaining 87 subjects with complete historical data constitute the `-obs` subset. This partitioning provides a high-fidelity setting that realistically simulates a common real-world scenario where new sensors are added to an existing network.

**BEAR**  The BEAR dataset contains complete HVAC data for 80 rooms (subjects) within a single building. We partition the subjects based on their spatial distribution. Specifically, from each floor, we randomly sample a mix of window-side and non-window-side rooms. This process yields an `-obs` subset of 29 subjects and a `-hid` subset of 10 subjects.

### A.5 SUMMARY OF THE COMPARISON BETWEEN FIDEL-TS AND PRIOR BENCHMARKS

We summarize the benchmark comparison into five aspects: Data Scale, Frequency, Variable Composition, and Leakage, to more intuitively highlight the advantages of our benchmark.

Table 9: Comparison between Fidel-TS and Prior Benchmarks

| Benchmarks | Data Scale | Frequency | Variable Composition | Leakage |
|---|---|---|---|---|
| Classic Unimodal | Small (most contains 10k points) | Moderate (e.g., hourly for ETT) | Vague | Pre-training |
| Classic Multimodal | Severe scarcity (Some only a few hundred points) | Low (most sampled weekly or monthly) | Vague | Pre-training Temporal Description |
| Ours | Large (millions of points) | High (5 minute-1 hour) | Clear (Subject and Channel) | – |

## B   FRAMEWORK IMPLEMENTATION

Our Universal Cross-modal Evaluation Framework supports both raw text and pre-computed embedding vectors while rigorously enforcing the temporal alignment between textual data and time series. To cater to the varied nature of modern models, our framework not only provides standard PyTorch (Paszke et al., 2019) interface, but also integrates HuggingFace Transformers (Wolf et al., 2020) for foundation models, and leverages PyTorch Lightning (Falcon et al., 2019) to accelerate the training of complex multimodal models. Recognizing the unique requirements of LLMs, it also supports both local deployment via vLLM (Kwon et al., 2023) and remote API calls through a simple socket. This thoughtful engineering ensures that our framework serves as the essential bridge between the rich, multimodal data of Fidel-TS and the diverse array of models, enabling the fair and comprehensive experiments that follow.

## C   MODEL DETAILS

Table 10 contains all the models we select for our experiments. We summarize their Training paradigm, Architecture, Modality and Textual Input Requirements. "General" and "channel/subject" corresponds to the dataset descriptions and the meanings of channels/subjects provided in the metadata. "History" denotes the textual information within the historical window. "Condition" refers to the conditions used for forecasting over the future horizon.

Note that LLM-based domain-specific models (GPT4TS, GPT4MTS) are distinct from LLMs. They only employ a frozen GPT-2 (Radford et al., 2004) as a backbone, and only the surrounding input

and output layers are trainable. Therefore, they must be trained on the dataset and cannot perform zero-shot forecasting.

Table 10: Key information of models

| Model | Training paradigm | Architecture | Modality | Textual Input Requirements |
|---|---|---|---|---|
| PatchTST | Domain-Specific | Num. trans. | Unimodal | — |
| iTransformer | Domain-Specific | Num. trans. | Unimodal | — |
| Dlinear | Domain-Specific | Linear | Unimodal | — |
| FITS | Domain-Specific | Linear | Unimodal | — |
| GPT4TS | Domain-Specific | LLM | Unimodal | — |
| Chronos | Cross-domain | Num. trans. | Unimodal | — |
| TimeMoE | Cross-domain | Num. trans. | Unimodal | — |
| Sundial | Cross-domain | Num. trans. | Unimodal | — |
| GPT4MTS | Domain-Specific | LLM | Multimodal | history |
| FIATS | Domain-Specific | Num. trans. | Multimodal | condition, general,subject/channel |
| Qwen2.5-14B-Instruct | Cross-domain | LLM | Unimodal/Multimodal | history, condition, general, subject/channel |
| Qwen2.5-14B-Instruct-1M | Cross-domain | LLM | Unimodal/Multimodal | history, condition, general, subject/channel |
| Qwen3-14B | Cross-domain | LLM | Unimodal/Multimodal | history, condition, general, subject/channel |
| Deepseek-R1 | Cross-domain | LLM | Unimodal/Multimodal | history, condition, general, subject/channel |
| Chattime | Cross-domain | LLM | Unimodal/Multimodal | condition, general, subject/channel |

Additionally, to ensure experimental transparency, we provide the prompt templates used for the unimodal and multimodal forecasting tasks with LLMs. The template for unimodal forecasting, illustrated in Figure 5 (a), incorporates only basic information about the dataset and channel, along with the formatted historical time series as input. The system prompt directly instructs the model to learn the pattern of time series and generate the forecast.

The multimodal forecasting template shown in Figure 5 (b) extends the unimodal setup by additionally providing historical textual information and future conditions in the prediction window. The corresponding system prompt directs the LLM to not only learn the pattern of time series but also focus on the association between the textual data and the time series. For both tasks, we explicitly define the output format to minimize the likelihood of formatting errors.

Notably, we design an additional prompt template for the multimodal forecasting of the BEAR dataset to accommodate its unique control events. The weather-related content is replaced with control-related information, while all other parts remained unchanged. We applied these templates only to native LLMs, including the Qwen series and Deepseek-R1. For the Chattime which fine-tuned Llama-2-7b, we utilized its original template.

## D EXPERIMENTAL SETTINGS DETAILS

In our experiments, we strictly follow to the principle of Realistic Settings. For all datasets except BEAR, we uniformly adopt the datasets downsampled to hourly (suffixed with -h). For these hourly datasets, we employ a historical horizon of 360 time steps (representing half a month) and prediction lengths of {24, 168, 336, 720}, corresponding to {1 day, 1 week, 2 weeks, and 1 month}, respectively.

The BEAR dataset, due to its limited time span, is not downsampled to prevent significant data loss. Consequently, we utilize its original 5-minute sampling frequency. The configuration for BEAR includes a historical horizon of 288 time steps (representing 1 day) and prediction lengths of {12, 144, 288, 576}, which correspond to {1 hour, 12 hours, 1 day, and 2 days}, respectively.

After selecting the appropriate version of each dataset, we partition them into training, validation, and testing sets. We employ the realistic split method based on timestamps. For all datasets except BEAR, we use {Jan. 1, 2021, Jan. 1, 2022}, as the split points to separate the three sets. For the BEAR dataset, the corresponding split points were set to {Sep. 1, 2020, Oct. 1, 2020}.

Specifically mentioned, for all domain-specific models, both unimodal and multimodal, training must be conducted on each dataset individually. We first train these models and subsequently test them on the full test sets. Following this, we apply our aforementioned importance sampling principle to construct the -mini test sets. For computationally expensive models such as LLMs, evaluation is performed exclusively on these lighter -mini test sets.

| <System>
You are a professonal data analyst. You can do the forecasting by only considering the historical time series and make the prediction for the following period of time. This dataset is about {dataset_info}. You can consider the periodicity, trend, and seasonality of the data. The time period are defined as follows, 0~6 as Early morning, 7~12 as Morning, 13~18 as Afternoon, 19~24 as Evening. Please directly do the forecasting based on your analysis and reasoning, don't use external tools or coding.
</System>
<User>
We now predict the sensor reading for {channel_info}. The historical data is as follows:
{historical_time_series}
Please format your forecasting in json format in [forecast_timestamp, predicted_value] pair. **Use the timestamp yyyymmddHHMMSS**. **The shape/length of the predicted_value must strictly match the shape/length of a data point in the historical data. If a historical data point is a single number, the predicted_value must also be a single number. If a historical data point is a list, the predicted_value must be a list of the exact same shape**. The predicted_value should be non-negtive float, it can be 0 when no data generate or the sensor/system is shutdown. Forecast 0 if the sensor is down, don't think too much about this. The json format is as follows with timestamp to forecast prefilled:
```json
{timestamps_for_prediction}
```
Now do the analysis step by step. Finally, MAKE SURE YOUR FINAL PREDICTION ALIGNS WITH YOUR ANALYSIS.
</User> | <System>
You are a professonal data analyst. You can do the forecasting by considering the historical time series readings along with weather condition and make the prediction for the following period of time. This dataset is about {dataset_info}. You can consider the periodicity, trend, and seasonality of the data. Then consider the affect of weather condition to further refine your prediction. The time period are defined as follows, 0~6 as Early morning, 7~12 as Morning, 13~18 as Afternoon, 19~24 as Evening. Please directly do the forecasting based on your analysis and reasoning, don't use external tools or coding.
</System>
<User>
We now predict the sensor reading for {channel_info}. The historical data is as follows:
{historical_time_series}
The historical weather condition is as follows:
{historical_textual_data}
We now have the weather forecasting for the following days:
{future_textual_data}
Please format your forecasting in json format in [forecast_timestamp, predicted_value] pair. **The shape/length of the predicted_value must strictly match the shape/length of a data point in the historical data. If a historical data point is a single number, the predicted_value must also be a single number. If a historical data point is a list, the predicted_value must be a list of the exact same shape**. **Use the timestamp yyyymmddHHMMSS**. The predicted_value should be non-negative float, it can be 0 when no data generate or the sensor/system is shutdown. Forecast 0 if the sensor is down, don't think too much about this. The json format is as follows with timestamp to forecast prefilled:
```json
{timestamps_for_prediction}
```
Now do the analysis step by step. Finally, MAKE SURE YOUR FINAL PREDICTION ALIGNS WITH YOUR ANALYSIS.
</User> |
|---|---|
| (a) Unimodal Forecasting | (b) Multimodal Forecasting |

Figure 5: Prompt template for LLM unimodal forecasting and multimodal forecasting

# E  ADDITIONAL EXPERIMENT RESULTS

## E.1  BENCHMARKING UNIMODAL FORECASTING WITHOUT DOWNSAMPLING

Table 11: Performance of unimodal forecasting models on full test set of each dataset without downsampling. All metrics are **MSE**.

| Dataset | Samp. Rate | His. Len. | Pred. Len. | Patch. | Dlinear | Sundial |
|---|---|---|---|---|---|---|
| GREG | 10min | 1440 | 96 | 0.091 | 0.106 | 0.097 |
|  |  |  | 672 | 0.131 | 0.154 | 0.166 |
| JAP | 15min | 2160 | 144 | 0.367 | 0.399 | 0.388 |
|  |  |  | 1008 | 0.557 | 0.565 | 0.610 |
| CAISO | 5min | 4320 | 288 | 0.185 | 0.191 | 0.299 |
|  |  |  | 2016 | 0.367 | 0.397 | 0.610 |
| NYTS-obs | 5min | 4320 | 288 | 0.600 | 0.618 | 0.844 |
|  |  |  | 2016 | 1.071 | 1.174 | 1.391 |

We evaluate the unimodal forecasting models on the datasets at their original sampling rates. To maintain the same forecasting setting (a half-month history to predict the next day, the next week, the next 2 weeks and the next month), we adapt the history and prediction lengths to match the native frequency of each dataset. Due to the prohibitive computational cost, we selectively conducted experiments on a subset of models over shorter prediction horizons (day and week), with the results presented in Table 11. These results indicate that the models' performance on the raw-resolution data is similar to that on the data downsampled to an hourly frequency.

Table 12: Performance of domain-specific multimodal forecasting models on full test set of each dataset. All metrics are **MSE**. We use control events for the BEAR dataset.

| Dataset | Pred. Len. | GPT4MTS | FIATS |
|---------|-----------|---------|-------|
| CCP-h | 24 | 0.162 | 0.109 |
|        | 168 | 0.212 | 0.159 |
|        | 336 | 0.226 | 0.168 |
|        | 720 | 0.236 | 0.183 |
| GREG-h | 24 | 0.093 | 0.044 |
|        | 168 | 0.139 | 0.065 |
|        | 336 | 0.151 | 0.082 |
|        | 720 | 0.169 | 0.094 |
| JAP-h | 24 | 0.358 | 0.333 |
|       | 168 | 0.543 | 0.451 |
|       | 336 | 0.605 | 0.500 |
|       | 720 | 0.656 | 0.502 |
| CAISO-h | 24 | 0.183 | 0.202 |
|         | 168 | 0.370 | 0.429 |
|         | 336 | 0.443 | 0.509 |
|         | 720 | 0.543 | 0.600 |
| NYTS-h-obs | 24 | 0.617 | 0.407 |
|            | 168 | 1.099 | 0.644 |
|            | 336 | 1.290 | 0.685 |
|            | 720 | 1.335 | 0.710 |
| BEAR-obs | 12 | 0.117 | 0.094 |
|          | 144 | 0.591 | 0.222 |
|          | 288 | 0.874 | 0.298 |
|          | 576 | 1.096 | 0.323 |

### E.2 FULL RESULTS FOR BENCHMARKING MULTIMODAL FORECASTING

Table 12 summarizes the performance of multi-modal, domain-specific models evaluated on the full test set of our benchmark. In a direct comparison between the two primary multi-modal approaches, FIATS demonstrates superior predictive accuracy over GPT4MTS. Furthermore, a cross-reference between Table 12 and Table 3 reveals distinct trends in multi-modal efficacy: FIATS outperforms uni-modal models in the majority of scenarios, validating the benefit of its multi-modal integration. Conversely, the marginal gains of GPT4MTS are limited; notably, it even exhibits performance degradation on certain datasets compared to its uni-modal counterpart GPT4TS.

### E.3 LLM REASONING EXAMPLES

To illustrate how LLMs utilize known information within the context in unimodal and multimodal forecasting scenarios, we compared the reasoning processes of Deepseek-R1 under both tasks. The examples we use are derived from the CCP-mini test set. The results are shown in Figure 6.

In the unimodal forecasting scenario, the model relies exclusively on historical time-series data and implicit domain knowledge. The reasoning process focuses on identifying periodicity (daily solar cycles) and seasonal trends (increasing daylight in the Northern Hemisphere). Crucially, the model explicitly acknowledges the lack of external variables, stating, "I don't have weather data." Consequently, it defaults to a statistical approach, assuming that recent weather will persist, and generates predictions by applying numerical multipliers derived solely from historical correlations.

In contrast, the multimodal forecasting scenario demonstrates a more robust reasoning capability by integrating textual weather data with numerical history. The model actively utilizes the provided weather forecast ("partly sunny skies") to select a specific reference day (March 10) that shares similar conditions. Instead of simple extrapolation, it employs a logic-driven adjustment: it calculates a ratio based on the difference between the current day and the reference day at a known timestamp (11 AM) and applies this scaling factor to forecast the subsequent hours. This highlights the model's ability to ground its predictions in semantic context rather than relying solely on historical data.

| Deepseek-R1 Unimodal Forecasting | Deepseek-R1 Multimodal Forecasting |
|---|---|
| <thinking...>
First, I need to forecast solar power generation for the next 24 hours... The historical data is provided, and I have to consider periodicity, trend, and seasonality. Since solar power is highly dependent on daylight, there should be a strong daily seasonality... From the historical data, I can see that solar power generation is zero during the night hours and peaks during the day, specifically in the afternoon... The historical data spans from February 25, 2024, to March 11, 2024... I should note that the data is from Calgary, Alberta, Canada, which is in the Northern Hemisphere, so in late February and early March, days are getting longer, so there might be a slight increasing trend in solar generation due to more daylight hours.
From the data, I can see that some days have high generation, and some have low, probably due to cloud cover or weather conditions... For example, on Feb 25... On Feb 26... On Feb 27... On Feb 28... On Feb 29... Then Mar 1: very low values, almost zero throughout the day, probably cloudy or snowy... So, from Mar 1 to Mar 7, generation was very low, likely due to poor weather, and it improved from Mar 8 onwards... For forecasting, I need to predict for the rest of Mar 11 and for Mar 12. **Since solar power is highly variable based on weather, and I don't have weather data, I need to rely on the historical pattern and recent trends...** From Mar 8, 9, 10, and 11 morning, generation has been good, so likely Mar 11 and Mar 12 will have similar good weather... So for Mar 11, I can use Mar 10 as a reference... For Mar 12, I need to forecast based on previous days... So for hours 19 to 23 and 0 to 6, I can forecast zero.
For Mar 11, at 11:00: 7.968, so for 12:00: 7.968 * 1.633 = 13.01... 13:00: 7.968 * 1.709 = 13.62... 14:00: 7.968 * 1.559 = 12.42... 15:00: 7.968 * 1.143 = 9.11... 16:00: 7.968 * 0.399 = 3.18... 17:00: 7.968 * 0.573 = 4.57... 18:00: 7.968 * 0.341 = 2.72... For Mar 12...
So for Mar 12: 7:00: 0.0, 8:00: 0.774, 9:00: 3.219, 10:00: 5.062, 11:00: 7.968...
Now, for the JSON output, I need to write the pairs. Now, let's write the forecast list.
<thinking end> | <thinking...>
First, I need to forecast solar power generation for the solar panel at Calgary Fire Hall Headquarters. The historical data is given in hourly readings, and I have weather conditions for past days and forecasts for the next couple of days. The timestamps for forecasting are from 20240311120000 to 20240312110000, which covers from March 11, 12:00 PM to March 12, 11:00 AM.
I need to consider the periodicity, trend, and seasonality of the solar power data. Solar power generation is highly dependent on daylight hours, so there should be a daily pattern where generation is zero at night and peaks during the day.
From the historical data, I can see that solar generation starts around sunrise, peaks in the afternoon, and drops to zero after sunset.
**Now, for the forecast period, I have weather forecasts for March 11 and March 12. From the weather forecast: For March 11: "partly sunny skies"... For March 12: "partly sunny skies"... March 10: high generation, weather: "partly sunny", so sunny. March 11: partly sunny, so similar to March 10. So the only fully sunny days with high generation are Feb 28, Feb 29, and March 10... So for March 11, I will use the same values as March 10 for the afternoon hours, adjusted for the slight difference at 11 AM.**
On March 10 at 11 AM: 7.52. On March 11 at 11 AM: 7.97, so ratio 7.97/7.52 = 1.06. So multiply March 10 afternoon values by 1.06. So for March 11: 12 PM: 12.27 * 1.06 = 13.01, 1 PM: 12.85 * 1.06 = 13.62...
But at 7 PM, it should be zero, so 0.07 is fine, and after that zero. Now for March 11, after 7 PM, generation should be zero, so for 8 PM to 11 PM, zero.
Now for March 12, we need to forecast from 12 AM to 11 AM. From 12 AM to 6 AM, early morning, no generation, so zero... Since weather is similar, I can use the same pattern as March 11 morning for March 12 morning... So for March 12: 7 AM: 0.0, 8 AM: 0.77, 9 AM: 3.22, 10 AM: 5.06, 11 AM: 7.97.
Now, for the values, I'll use the above. Now, I think this is reasonable. So let's write the JSON.
<thinking end> |

Figure 6: Comparison of the Reasoning Processes of Deepseek-R1 on Unimodal and Multimodal Forecasting Tasks

