# OpenReview forum: "Fidel-TS: A High-Fidelity Benchmark for Multimodal Time Series Forecasting"
_ICLR.cc/2026/Conference — Submitted to ICLR 2026_

### Official Review · Reviewer_Jj4v · 2025-10-20

**Soundness:** 3
**Presentation:** 3
**Contribution:** 3
**Rating:** 6
**Confidence:** 4

**Summary:**

This paper introduces Fidele-TS, a new high-fidelity benchmark for multimodal time series forecasting. It addresses critical shortcomings in existing benchmarks, which often suffer from pre-training data contamination, causal and description leakage, and outdated or small-scale datasets that create an illusion of progress. The authors formalize key principles of high-fidelity benchmarking: (1) data sourcing integrity using live, authentication-protected APIs to ensure freshness and prevent contamination; (2) strict causal soundness by incorporating only exogenous textual information such as weather forecasts; and (3) structural clarity through clear separation of forecasting subjects and data channels. Built upon these principles, FIDEL-TS provides millions of high-frequency, leak-free data points with aligned textual information. Extensive experiments show that prior benchmarks overestimated model capabilities, and that genuine multimodal performance gains depend on the causal relevance of textual data. This benchmark establishes a robust and causally sound foundation for evaluating modern forecasting models.

**Strengths:**

1. The paper tackles a crucial gap in time-series forecasting by addressing benchmark contamination and data leakage, providing a strong conceptual and empirical foundation for fair evaluation.
2. FIDEL-TS is built on clear, rigorous principles—data integrity, causal soundness, and structural clarity—ensuring high realism and reproducibility.
3. Extensive experiments across diverse models and datasets convincingly demonstrate the benchmark’s validity and reveal hidden biases in prior evaluations.

**Weaknesses:**

1. The proposed benchmark heavily depends on live API data sources, which may challenge long-term reproducibility and accessibility.
2. The causal soundness principle, though central, lacks a formal quantitative validation or theoretical grounding beyond qualitative reasoning.
3. Experimental comparisons focus mainly on benchmark-driven insights, with limited exploration of computational efficiency, sensitivity analyses, or detailed ablation studies that could further strengthen methodological rigor.

**Questions:**

See weaknesses

---

> ### Author Response · Authors · 2025-11-21
> **Response to Reviewer Jj4v (Part 1)**
>
> We are very grateful to your positive assessment and insightful feedback. We are particularly encouraged that you found our work to be a "strong conceptual and empirical foundation for fair evaluation" and that you recognized the value of our "clear, rigorous principles."
>
> **1. Response to weakness 1**
>
> Our decision to rely on live APIs was a deliberate and essential choice, driven by the core goals of our work. As you noted in the strengths, this approach is fundamental to guaranteeing data freshness and, most critically, to creating a strict separation from the static web crawls that form the pre-training corpora for Large Language Models. This is our primary defense against the data contamination and leakage that plague existing benchmarks.
>
> While we acknowledge the theoretical risk associated with reliance on external services, we believe this concern is minimal in practice for several reasons:
>
> * **Stability and Official Backing**: The APIs we selected are not ephemeral or niche data sources. They have been operating stably for many years (for most of selected APIs, over a decade) and are widely accessed by both academic and commercial entities. Furthermore, several are maintained by official or governmental bodies (e.g., the NYC traffic data), which provides strong assurance of their long-term operational reliability.
> * **Benchmark snapshots**: Furthermore, to mitigate the failure risk and ensure long-term reproducibility, we can provide periodic (e.g., annual) static snapshots of the collected data on our project page, while still emphasizing the live APl for evaluation.
> * **Robustness of our pipeline**: If the current APIs we used unfortunately fails, we have many alternatives to switch to, which enhances the robustness of our pipeline. For time series data in different domains, we can utilize APIs from other sources. Additionally, there are numerous alternative API sources for weather reports worldwide, so obtaining weather text is not an issue.
>
> **2. Response to weakness 2**
>
> We believe the point you raised is valid in principle, but it also has its limitations when applied to our context. In an ideal setting, a theoretical and quantitative analysis of causal relationships would indeed be the most rigorous approach. However, **such formal analysis becomes exceedingly challenging, if not intractable, in the complex, real-world scenarios that our benchmark is designed to model.** The datasets we use are sourced from operational systems like power grids and traffic networks. These are not idealized environments suitable for controlled causal analysis; they are influenced by countless unobserved confounders and stochastic dynamics.
>
> Our central goal, as you identified, is to ensure the causal soundness principle is upheld. To achieve this, **we relied on strong, universally accepted common-sense priors about the world**. For example, it is a matter of established real-world knowledge that weather is a significant causal driver for many systems (like power grids and traffic), while the state of those systems does not, in turn, have a meaningful impact on meteorological patterns. By selecting weather as a universal source of textual input, we are leveraging this clear, unidirectional causal relationship to create forecasting scenarios that are grounded in reality and free from the confounding leakages present in prior benchmarks. Our approach is to build the benchmark on causal relationships that are so evident they do not require complex discovery algorithms to validate.
>
> For researchers interested in the formal theoretical aspects of this topic, we would enthusiastically recommend the rich body of work in the field of causal inference, as well as emerging research that specifically combines causal discovery with time series forecasting.

---

> ### Author Response · Authors · 2025-11-21
> **Response to Reviewer Jj4v (Part 2)**
>
> **3. Response to weakness 3**
>
> We believe the **computational cost** is justified, as it can reveal differences in computational efficiency during inference among different types of models. We commit to including model parameter counts in Appendix of the paper and will analyze their computational costs. Additionally, a key focus of our study is the failure cases of LLMs, which refers to situations where LLMs do not strictly adhere to the prompts and fail to output predictions in the specified format (Section 4.4, table 4 and 5). We used the pass@k (k=3) metric to evaluate, which measures the proportion of samples for which the LLM successfully completes the forecasting task within 3 given attempts. Common failures include producing predictions shorter than the required length or incorrectly identifying the number of variables, leading to wrong channel counts.
>
> While in-depth **analyses of sensitivity** or **model-specific ablation studies** are undoubtedly important, they are typically the centerpiece of papers that propose a new model architecture or methodology. The primary goal of our experimental section, however, was to focus on benchmark-driven insights, as we believe this is the most critical contribution for a paper introducing a new evaluation standard.
>
> As you astutely pointed out in your first strength, our paper's main objective is to "tackle a crucial gap in time-series forecasting... providing a strong conceptual and empirical foundation for fair evaluation." To fulfill this objective, our experiments were specifically designed to achieve what you highlighted in your third strength: to "convincingly demonstrate the benchmark’s validity and reveal hidden biases in prior evaluations." Our focus was therefore on using a diverse suite of established models to stress-test the benchmark itself and to reveal the landscape of model performance in a leak-free, realistic environment.

---

> > ### Comment · Reviewer_Jj4v · 2025-11-27
> >
> > I think my concerns are addressed by the author's justification.

---

> ### Author Response · Authors · 2025-11-26
> **Paper revision**
>
> Dear Reviewer,
>
> We have incorporated all the modifications committed to in our earlier response, and these changes are marked in blue in the revised manuscript. Regarding the specific questions you raised, we have summarized the details below for your convenience:
>
> 1. We updated the organization of the Experiment section, added a new Discussion section (Section 5), and moved the LLM reliability (pass rate) analysis to Section 5.1 (lines 447-474). Addtionally, we also added a computational cost analysis in Section 5.1 (lines 475-489).
> 2. We added an analysis of the llm reasoning process example in Appendix E.3 (lines lines 1061-1133). We illustrated the differences in the way LLMs think due to the varying known information in the context under unimodal and multimodal inputs.
>
> Thank you very much for your valuable comments and suggestions.

---

### Official Review · Reviewer_1Gia · 2025-10-30

**Soundness:** 1
**Presentation:** 1
**Contribution:** 2
**Rating:** 2
**Confidence:** 4

**Summary:**

The authors propose a new benchmark for multimodal forecasting, where the authors claim to address the issues of “data sourcing integrity”, “causal soundness” and “structural clarity” with prior benchmarks. Their proposed benchmark, Fidel-TS sources target time series from live APIs. They test a variety of models (unimodal models, multimodal models, LLMs) on this benchmark.

**Strengths:**

- The design of the benchmark is well thought-out. I like how the authors differentiate between the target time series and the context time series.
- The idea behind using real-world sources for both the time series and textual data seems interesting, and the authors present useful ideas in the paper to build real-world benchmarks for the multimodal forecasting task.

**Weaknesses:**

- **Unable to understand the choice behind weather being the primary textual modality**: The authors state that they focus on weather as the primary textual modality. How do the authors know that the weather affects the variables in an observable way? i.e. weather may affect traffic speed, photovoltaics etc. but is the effect visually visible in the time series and not confounded with other variables that might make the effect unobservable in the time series? We need to remember we are dealing with open systems with unobserved confounders etc. so we need to take all this into account when building such a benchmark.
I am aware this is a first step towards that, but the authors should verify
(1) Context is necessary to predict the target time series (See CiK [1] for more details on why this is important). If this is not verified, then it brings into question the quality of the benchmark.
(2) Verify whether changes in weather are correlated with changes in the target time series, and that the correlations are visible.
(3) If possible, make a subset of windows in the target time series which require the context more than the other windows. Surely, not all the windows in the target time series would require the context for prediction. Marking such windows would allow one to evaluate models separately on both subsets of windows.
Importantly, I believe that this verification should not be done with a model that they train on the data (which will be trained to use the text signals) but manually with humans or an alternate independent mechanism.
I’ve re-read the paper several times and I do not think either of the first two points have been ensured.

- **Figure 2 is confusing and completely different from what the authors discuss in the main text**: the authors talk about a dynamic event database in figure 2, and how there is an “Event list” etc. but there is no mention of any of these terms in the main text. There is a huge discrepancy between the figure and the text.

- **Extremely poor presentation**: It was extremely difficult for me to read the paper. There is excessive unnecessary text that severely interrupt the flow of the text. There are too many instances of these for me to point out. There are also too many passages in the text that seem to be heavily LLM-written.
(Note that I’ve compared this paper to other papers in my batch, and it was the one with the least understandable presentation)

- **Unable to understand the intuition behind Table 2**: Why does the table solely show the performance of unimodal models, on a benchmark meant for multimodal forecasting? What is the purpose of these results? If it is to purely benchmark the models, unimodal results should only be used for comparison with multimodal results, as by themselves the unimodal results do not have any value.

- **Unable to reason why testing LLMs zero-shot on this setup makes sense**: It does not make sense for me to test LLMs zero-shot on this benchmark. Clearly, there are two different kinds of benchmark for multimodal forecasting according to [2]: training-based (Time-MMD [3], ChatTime [5]) where models require training to understand the nuances of the data in the benchmark, and zero-shot (CiK [1]) where context-aided forecasting only requires the model to apply a clear scenario on a forecast (as explored also in [4]) This benchmark clearly falls under training-based, meaning that models should be trained to use the correlations between the context and the time series. It doesn't make sense to directly test LLMs on such data that requires training to be used.

- **Insufficient evaluation of LLMs in the zero-shot setup**: If I am correct, the authors only evaluate LLMs with a prompting methodology similar to Direct Prompting as presented in CiK [1]. However this is not the only prompting methodology to use LLMs for forecasting; there also exists LLMP / LLMTime (CiK evaluates the LLMP methodology) and it has been shown in CiK that the strategy clearly makes a difference. The authors must evaluate these strategies as well, at least with a subset of LLMs.

- **Unfair comparison of LLMs with trained multimodal models**: Foregoing the last point, comparing LLMs zero-shot and claiming that they are worse than the multimodal models is not fair; when the other multimodal models are trained on a subset of the data. Please either reword the claims or mention the discrepancy in the setup.

- **Unable to understand how the benchmark is different from the others**: It is still unclear how the benchmark is different from the other benchmarks. I’d prefer to look at a table which clearly shows how this benchmark differs.

- **Confusing experiment results**: The authors show that the FITS model gives worse results when weather is incorporated; however in the benchmark the authors do not mark any of such cases and this is purely an empirical results. The authors do not also claim that benchmark has such “bad actors” in the context, and instead if I’m right, they only claim that context is relevant and the benchmark is useful to test multimodal forecasting capabilities.
   - These results make me re-think the quality of the entire paper, as such instances may have been observed in all models, but with the conclusion that the model cannot use context.
   - **Suggestion**: Either mark or remove the bad actor variables so we can appropriately evaluate models without them

- **Limited Analysis of the results**: Adding to the above point, the authors present a very limited analysis of results:  no examples are provided of the forecasts of LLMs with and without context, demonstrating how the context meaningfully changes the forecasts.

- **No discussion on how the different models compare in terms of cost and parameter count**: An issue that the CiK paper [1] highlighted is the high cost of LLMs. That is not discussed here. I’m not sure how comparable the different LLMs and multimodal models are, in terms of parameter count.

## Minor

- **Spelling mistakes (minor)**: The authors consistently use “FIATS” in Section 4.4 - I think they are referring to the FITS model but this should be corrected. This is repeated again and again; I don’t think this is professional on the authors’ part to not proofread their paper for such mistakes.


- **Unnecessary claims on implementation**: “To cater to the varied nature of modern models, our framework not only provides standard PyTorch (Paszke et al., 2019) interface, but also integrates HuggingFace Transformers (Wolf et al., 2020) for foundation models, and leverages PyTorch Lightning (Falcon et al., 2019) to accelerate the training of complex multimodal models. Recognizing the unique requirements of LLMs, it also supports both local deployment via vLLM (Kwon et al., 2023) and remote API calls through a simple socket”
I do not see any of this as novelty, and worth mentioning in the main text at all.
		The authors only propose a time series x text benchmark. All tested models are from other papers, which are already based on PyTorch/HuggingFace etc.


### Summary Note:
The only reason for me giving a Score of 2 is I see some value in the benchmark that the authors are attempting to build. However if they are addressing the mentioned issues, they should address it properly in a principled way, and further evaluate models the correct way, and make the right claims. The presentation is completely off (thereby a score of 1 from me), as per the paper the approach is not sound at all (thereby a score of 1 again).

**Questions:**

- **“Strict Causal Soundness”**: The authors state “Strict Causal Soundness: which incorporates only verifiably exogenous textual information, such as weather forecasts and scheduled maintenance, to prevent causal and description leakage”
    - However, I feel the term “Strict Causal Soundness” doesn’t make sense for what the authors describe. Causal soundness would ideally refer to how the variables in the context are causal parents of the target time series variables, which the authors do not verify. I would rather the authors term this “External Variables” or something like that to not mislead the readers and make the wrong claims.
- **“Causal leakage”**: The same with “Causal leakage” which the authors term as “retrieved documents contain future information unavailable at prediction time” which is just temporal leakage. Any claims of causality should be carefully made. I suggest the authors reword this.
- **“Ambiguous variable structure”**: The authors discuss an “ambiguous variable structure” - I don’t think this is a problem at all, as the referred unimodal benchmarks are only meant for univariate forecasting, meaning there is no variable structure. Therefore this is only an issue when it comes to covariate-informed forecasting or multimodal forecasting. But the authors say that “it perpetuates the ambiguous variable structure of classic unimodal benchmarks” which doesn’t make sense.
Question on wording: “To ensure our evaluation mirrors the practical scenario of forecasting, we extend our high-fidelity principle from the benchmark to the experimental setup itself.”
I’m not sure what the authors mean here. Can you clarify?

To note, all my questions mostly point to the poor presentation of the work. I suggest the authors carefully understand this and ensure that their contributions are clearly communicated to readers.

## REFERENCES

[1] Williams, Andrew Robert, Arjun Ashok, Étienne Marcotte, Valentina Zantedeschi, Jithendaraa Subramanian, Roland Riachi, James Requeima et al. "Context is key: A benchmark for forecasting with essential textual information." ICML 2025.

[2] Zhang, Xiyuan, Boran Han, Haoyang Fang, Abdul Fatir Ansari, Shuai Zhang, Danielle C. Maddix, Cuixiong Hu et al. "Does Multimodality Lead to Better Time Series Forecasting?." arXiv preprint arXiv:2506.21611 (2025).

[3] Liu, Haoxin, Shangqing Xu, Zhiyuan Zhao, Lingkai Kong, Harshavardhan Prabhakar Kamarthi, Aditya Sasanur, Megha Sharma et al. "Time-mmd: Multi-domain multimodal dataset for time series analysis." NeurIPS 2024.

[4] Ashok, Arjun, Andrew Robert Williams, Vincent Zhihao Zheng, Irina Rish, Nicolas Chapados, Étienne Marcotte, Valentina Zantedeschi, and Alexandre Drouin. "Beyond Na\" ive Prompting: Strategies for Improved Zero-shot Context-aided Forecasting with LLMs." arXiv preprint arXiv:2508.09904 (2025).

[5] Wang, Chengsen, Qi Qi, Jingyu Wang, Haifeng Sun, Zirui Zhuang, Jinming Wu, Lei Zhang, and Jianxin Liao. "Chattime: A unified multimodal time series foundation model bridging numerical and textual data." In Proceedings of the AAAI Conference on Artificial Intelligence, vol. 39, no. 12, pp. 12694-12702. 2025.

---

> ### Author Response · Authors · 2025-11-21
> **Response to Reviewer 1Gia (Part 1)**
>
> We thank you for your time and for providing detailed feedback on our manuscript. We would like to take this opportunity to clarify several points of potential misunderstanding regarding our motivation, experiments, and presentations. We have summarized your question into three main points and will reply to each one as follows.
>
> **1. Response to your questions about our motivation and benchmark design**
>
> **1.1 Context is necessary for multimodal time series forecasting**
>
> As we outline in our introduction (Lines 026-032), the field of time series forecasting has a long history, traditionally focusing on unimodal models that learn from historical numerical data alone. The recent rise of multimodal forecasting is driven by the fundamental real-world principle that external context significantly impacts system dynamics. **Recent work has empirically validated that incorporating relevant textual information can substantially boost forecasting performance.** For instance, the theoretical analysis in work Intervention-Aware Forecasting (Xu et al., 2024) suggests that while unimodal forecasting has a performance ceiling, multimodal approaches can break past this limit by leveraging richer contextual signals. Other representative works such as context is key (Williams et al., 2024), from news to forecast (Wang et al., 2024) have also demonstrated this in various ways. The TimeMMD benchmark (Liu et al., 2024a) was a pioneering first step in this direction, providing a dataset for this emerging subfield.
>
> **1.2 The unique value of our benchmark (compared with previous benchmarks)**
>
> Our work is motivated by a critical need to **address foundational issues in both unimodal and multimodal benchmarks**, especially when unimodal and multimodal time series forecasting models are developing rapidly. As we argued in our paper (Section 2, lines 72-121), the uniqueness of our benchmark compared to the previous ones lies in two aspects:
> * **Compared to Classic Unimodal Benchmarks**: These datasets often suffer from being small-scale, outdated, low-frequency and ambiguous variable structure (Lines 033-036).
> * **Compared to the First Multimodal Benchmark**: While TimeMMD laudably incorporated diverse text sources like news and economic reports, this very breadth introduced severe flaws. As we detail in our paper (Lines 041-045), these include pre-training contamination, temporal leakage, and description leakage. Evaluating models on a benchmark with these issues creates an artificial test scenario that does not reflect real-world forecasting constraints. This is precisely why we made the **deliberate choice to begin with a carefully selected source "weather".** In addition, TimeMMD also has the problems of low sampling frequency and ambiguous variable structure, which is the same as unimodal benchmarks.
> * **Regarding your specific question of "ambiguous variable structure"**: The domain of our paper is Long-Term Time Series Forecasting. In standard LTSF tasks, the default setting is to use a multivariate history window to forecast a multivariate prediction window. This process inherently involves a variate structure within the time series.
> However, a critical oversight in all prior benchmarks has been **the failure to distinguish between two key concepts: subjects and channels**. In the context of multivariate forecasting, previous datasets did not explicitly differentiate whether the multiple variables originated from the same channel across multiple subjects, or from multiple channels of a single subject (As we argued in Section 3.1, lines 161-184).
> **Therefore, the univariate/multivariate variate structure we refer to is an intrinsic property of the time series data within the benchmark itself.** This concept is distinct from, and should not be confused with, unimodal/multimodal learning or covariate-informed forecasting.
>
> We will organize these main differences into a table and add it to our Appendix:
>
> | Dataset | Data Scale | Frequency | Variable Composition | Modality | Fidelity |
> | :--- | :--- | :--- | :--- | :--- | :--- |
> | **Classic Unimodal Benchmarks** | Small (most contains 10k points) | Moderate (e.g., hourly for ETT) | Vague | Unimodal | Low |
> | **Classic Multimodal Benchmarks** | Severe scarcity (many series have only a few hundred points) | Low (most sampled weekly or monthly) | Vague | Multimodal | Low |
> | **Ours** | Large (provides millions of points) | High (5 minite-1 hour) | Clear (distinguish between subject and channel) | Multimodal | High |

---

> ### Author Response · Authors · 2025-11-21
> **Response to Reviewer 1Gia (Part 2)**
>
> **1.3 The reasonability of the weather**
>
> For an idealized system, it is indeed possible to theoretically demonstrate a causal link between weather and system variables, justifying weather as a textual input. However, our benchmark is specifically designed to address real-world systems, which are inherently complex and influenced by innumerable confounding factors. In such non-ideal scenarios, providing a rigorous theoretical proof of causality is exceptionally difficult, if not impossible. But as we argue in our manuscript (lines 144-153), we adopted a pragmatic approach **grounded in real-world common sense, which is both reasonable and easily verifiable**. Weather acts as a near-universal exogenous factor that influences the majority of real-world systems (e.g., power, traffic). Crucially, this influence is unidirectional: weather affects these systems, but the state of these systems does not affect the weather. At the same time, this choice perfectly aligns with our core principle of real-world fidelity: (1) Weather forecasts are available in advance, mimicking a realistic forecasting pipeline. (2) The text does not directly describe the state changes of the target variables.
>
> **2. Response to your questions about our framework and experimental setups**
>
> **2.1 Framework Implementation**
>
> The models evaluated in our work are sourced from their original papers, which are accompanied by their official code repositories and distinct implementations. However, **a key limitation is the lack of a unified framework capable of accommodating these disparate implementations.** Therefore, our contribution here is the development of a unified evaluation framework. This framework enables users to run various types of models from different implementations within a single environment, simply by modifying configuration scripts. This design facilitates the evaluation of all models under a consistent set of configurations, ensuring a fair and direct comparison.
>
> **2.2 The principle of High-Fidelity experimental setup**
>
> The principle of high-fidelity experimental setup refers to the alignment of the history and prediction window lengths with their real-world temporal significance. Previous long-term forecasting tasks often overlooked this principle, which is evident in two main aspects:
> (1) Both the history and prediction window lengths were fixed values, disregarding the real-world sampling frequency of the time series.
> (2) Dataset splitting was performed solely based on fixed ratios (e.g., 7:1:2).
> In contrast, when using our benchmark, **we advocate for setting history and prediction lengths that are more tightly coupled with real-world time**. For example, on datasets uniformly sampled at an hourly rate, the input and output lengths directly correspond to specific real-world time spans. Furthermore, our dataset splitting can be based on timestamps rather than fixed ratios.

---

> ### Author Response · Authors · 2025-11-21
> **Response to Reviewer 1Gia (Part 3)**
>
> **3. Response to your questions about our experiment design and results**
>
> **3.1 The FIATS model**
>
> FITS and FIATS are two entirely different models from different papers.
>
> * FITS [1] is a unimodal forecasting model.
> * FIATS [2] is a multimodal forecasting model.
>
> **We have cited FIATS correctly in our paper. This is not an unprofessional spelling mistake but a distinction between two separate lines of work**.
>
> [1] Xu, Z., Zeng, A., & Xu, Q. FITS: Modeling Time Series with $10 k $ Parameters. In The Twelfth International Conference on Learning Representations.
>
> [2] Xu, Z., Wang, H., & Xu, Q. (2024). Intervention-Aware Forecasting: Breaking Historical Limits from a System Perspective. arXiv preprint arXiv:2405.13522.
>
> **3.2 The value of table 2**
>
> As we state in our motivation, our dataset is designed to address shortcomings in **both existing unimodal and multimodal benchmarks**. So the first part of our experiments (Section 4.3, lines 335-343) focuses on the unimodal aspect. We use our large-scale and newly-collected dataset to evaluate a range of unimodal models, observing their rankings and respective performance. **The significance of this experiment is to reveal the biases introduced by older benchmarks.** It demonstrates that the performance of unimodal models is highly sensitive to the benchmark on which they are evaluated.
>
> **3.3 LLM zero-shot testing**
>
> The domain of our entire paper is Long-Term Time Series Forecasting, and we are proposing a benchmark specifically for this task. Whether a benchmark can be used for zero-shot testing does not depend on whether it is a so-called "training-based dataset." From our perspective, the key criterion is the model's suitability for the task.
> It is a common practice and a standard protocol within the time series forecasting community to evaluate Time Series Foundation Models (TSFMs) and Large Language Models (LLMs) in a zero-shot setting. The very design philosophy behind these models is to train larger, more general-purpose models whose zero-shot forecasting capabilities can be on par with domain-specific models that require explicit training. W**e argue that the comparison between zero-shot and training-based approaches is not inherently unfair, but rather reflects their distinct characteristics and the different paradigms under which each claims state-of-the-art (SOTA) performance.**
> As we demonstrate in Appendix C (Table 6), we have strictly followed the original configurations proposed in the papers for all models (whether they are unimodal or multimodal, trainable or non-trainable), providing each with its most suitable input. Specifically for LLMs, we designed tailored prompts (Figures 4 and 5) to **create distinct scenarios** for both unimodal and multimodal forecasting:
> * **Unimodal Scenario**: We provide the LLM with only the historical time series, prompting it to discover underlying patterns from the data itself.
> * **Multimodal Scenario**: We provide the LLM with the historical time series along with both historical and future weather reports. As analyzed based on real-world operational knowledge, weather serves as a competent auxiliary context for prediction. This is precisely why we designed this scenario for LLMs.
>
> Therefore, our benchmark is designed to be versatile. Its test set can be used independently for zero-shot evaluation, and alternatively, the training and validation sets can be used to train domain-specific models before evaluation on the test set. It is fully compatible with both paradigms.
>
> **3.4 Model computational cost and parameter counts**
>
> The parameter counts of LLMs are easily accessible (e.g., the larger Deepseek-R1 at 671B, or the smaller Qwen-2.5 at 14B). In contrast, domain-specific multimodal forecasting models that require training have significantly fewer parameters (e.g., FIATS with 11M).
> We acknowledge the value of this information. Therefore, we will add the parameter counts for all models to our model list (in Appendix) and provide a brief analysis of the computational cost. This analysis will highlight that the inference cost of LLMs is higher than that of the trainable, domain-specific models.
>
> Distinct from the discussion of computational cost, our paper does address another critical issue: the stability of LLMs. We reported the pass rate of test samples over three attempts, pass@k (with k=3), for LLMs to observe potential failure cases where the LLM does not adhere to the prompt and produces incorrectly formatted output. The primary point we emphasize is the immaturity of using direct prompting for forecasting tasks and the inherent instruction-following limitations of current LLMs.

---

> ### Author Response · Authors · 2025-11-21
> **Response to Reviewer 1Gia (Part 4)**
>
> **3.5 Examples of LLM outputs**
>
> This is indeed a valuable discussion that is currently missing from our paper. We commit to provide examples and illustrate the model's behavior in both scenarios (with/without textual data) to provide a clearer picture of its capabilities and limitations.
>
> **3.6 LLM evaluation strategies**
>
> We acknowledge the existence of alternative prompting strategies suchas LLMP/LLMTime, but this goes beyond the main goal of our benchmark-driven experiment. Our primary goal was to establish baseline LLM performance using the most common approach (Direct Prompting). Given the significant computational cost, and the already poor performance and stability issues (low pass rates) observed with direct prompting on our benchmark, we prioritized model breadth over the depth of prompting strategies.
>
> The different prompting methods you mentioned are indeed a meaningful evaluation that can be carried out on our benchmark. Due to the large parameter scale of some LLMs and the limitation of our computing resources, we adopted the evaluation method of calling APIs through LLM sockets. It is difficult to directly implement complex strategies (LLMP/LLMTime) under this architecture. We will note this as a limitation and encourage the exploration of other prompting techniques on Fidel-TS as valuable future work.
>
> **3.7 "Bad Case" of weather context on the BEAR dataset**
>
> Our intention is not to suggest that weather data is an inadequate source or that its inclusion diminishes the quality of a benchmark. On the contrary, we recognize its limitations: while weather is a ubiquitous factor influencing most real-world systems, it may not always be the most causally relevant one.
>
> Our work is fundamentally constrained by the principle of maintaining real-world fidelity. With this in mind, we sought to identify a text source that is both true to the real world and more directly impactful on the system's dynamics than general weather patterns. This led us to augment the BEAR dataset with control information on planned maintenance. Therefore, the primary objective of this experiment was to provide a concrete demonstration of our proposed "causal soundness principle". We have made exolaination in our paper (Section 4.4, lines 440-463).
>
> This exploration directly informs our future work: to continue expanding beyond weather to incorporate more domain-specific, causally sound factors, all while adhering to the principle of real-world fidelity. It also points toward our goal of designing a pipeline to filter and select optimal textual data for forecasting. We understand that this experiment may have caused some confusion, so we will stress the following conclusion for readers in our paper: for the highest quality of evaluation, the control event texts from the BEAR dataset should be used.
>
> **4. Response to your questions about the presentation of our papers**
>
> **4.1 Paper writing**
>
> We tried to structure our paper to follow a logical progression. As disclosed in our LLM Usage statement, while we utilized a large language model for polishing the text, we did not use it to directly generate substantial sections of content. Although you did not provide specific examples, citing that the issues were too numerous, we will conduct another thorough review of the entire manuscript and revise any passages that may be unclear or awkwardly phrased to improve overall readability.
>
> **4.2 The use of "causal leakage"**
>
> We will replace "causal leakage" with "temporal leakage" to avoid potential misunderstanding.
>
> **4.3 Figure 1**
>
> We provides a general overview of our benchmark construction pipeline in Figure 1. The "Dynamic Event Database" is the source of raw data for the text inputs, such as weather and planned control events we proposed. To eliminate any confusion, we will amend the accompanying text to ensure it perfectly matches the diagram's components and flow.

---

> ### Author Response · Authors · 2025-11-26
> **Paper revision**
>
> Dear Reviewer,
>
> We have incorporated all the modifications committed to in our earlier response, and these changes are marked in blue in the revised manuscript. Regarding the specific questions you raised, we have summarized the details below for your convenience:
>
> **About Motivation and design**
>
> 1. We provided a clearer explanation of why we deliberately chose weather in Section 3.1 (lines 140-150)
> 2. We added a more intuitive comparison table between our benchmark and the previous benchmarks in Appendix A.5 (lines 883-895)
>
> **About Framework**
>
> 1. We moved the framework comparison from the original Appendix B to the main text (Section 4.1, lines 254-279), which more clearly illustrates our contribution. The implementation details are placed in Appendix B (lines 897-908)
>
> **Abput Presentation**
>
> 1. We changed "causal leakage" to "temporal leakage" and "strict causal soundness" to "leak-free and causally sound design" to avoid misleading.
> 2. We rewrote the caption of Figure 1 and the text introducing the text construction process, aligning the key parts in the figure with the text
> 3. We updated the organization of the Experiment section, added a new Discussion section (Section 5). The specific changes will be introduced in the "About Experiments" part.
>
> **About Experiments**
> 1. We moved the LLM reliability (pass rate) analysis to Section 5.1 (lines 447-474). Addtionally, we also added a computational cost analysis in Section 5.1 (lines 475-489).
> 2. We added an analysis of the llm reasoning process example in Appendix E.3 (lines lines 1061-1133). We illustrated the differences in the way LLMs think due to the varying known information in the context under unimodal and multimodal inputs.
> 3. In Section 5.2, we elaborated on the ablation study and clearly concluded that "for the BEAR dataset, more causally sound control events outperform weather conditions". This leads to our limitations and future works In Section 5.3 (lines 507-522).
> 4. We also acknowledged the limitation of lacking detailed comparison of specific methods (LLMP/LLMTime) due to the diversity of models and the limitations of computing resources. Future works will involve evaluating more models or methods (Section 5.3 (lines 507-522)).
>
> Thank you very much for your valuable comments and suggestions.

---

### Official Review · Reviewer_xbyV · 2025-11-02

**Soundness:** 3
**Presentation:** 3
**Contribution:** 4
**Rating:** 6
**Confidence:** 4

**Summary:**

The paper introduces a new benchmark for text+time-series multimodal forecasting which overcomes existing time-series datasets. Prior benchmarks suffer from pre-training data contamination, causal and description leakage. Fidel-TS is designed on: data sourcing integrity (using live, authentication-protected APIs), strict causal soundness (including only exogenous, verifiable textual data to avoid leakage), and clear separation of “subjects” and “channels” to assess model generalization. The benchmark includes multiple high-frequency datasets with aligned textual data such as weather forecasts and control events. Experiments using various unimodal, multimodal, and large language models (LLMs) show that previous benchmarks inflated model performance due to leakage.

**Strengths:**

1. Well motivated benchmark design: Introduces clear, principled criteria for dataset integrity, causal soundness, and structure.
2. Use of live API streams prevents contamination and ensures temporal realism.
3. Evaluates a wide range of models and exposes limitations in prior benchmarks.
4. Provides a scalable reproducible framework allowing future integration of new modalities and data sources.

**Weaknesses:**

1. Focuses mainly on weather and control events. Can you add more complex domains like news or economics?
2. Evaluation scope: While comprehensive, some comparisons (EG: non-live alternative data sources) could be further expanded.
3. Evaluation on long context inputs long horizon tasks. Critical vs simple data domains; adapting to sudden distributional shifts in temporal domains etc.

**Questions:**

See Weaknesses

---

> ### Author Response · Authors · 2025-11-21
> **Response to Reviewer xbyV (Part 1)**
>
> Thank you for your thoughtful review. We are grateful that you found our benchmark design to be well-motivated and supported that it successfully addresses structural issues in existing datasets. Our understanding aligns perfectly with one of our core objectives.
>
> **1. Response to weakness 1**
>
> Beyond the structural improvements, another primary motivation for our benchmark design is to address the pervasive issue of text data leakage in multimodal time-series forecasting (as discussed in Section 2.2, lines 94-120). **We argue that the immediate priority for the community should not be to indiscriminately expand the variety of text sources, but rather to first establish and adhere to the principle of real-world fidelity.** This principle guided our decision to primarily use weather data as the textual input. **This choice consciously avoids potentially confounding information, such as economic report or news**, which may not be available in a realistic, real-time forecasting scenario and thus "leaks" future information.
>
> As we acknowledge in our "Limitations and Future Works" section (lines 477-485)., we fully agree that expanding the benchmark with more domain-specific text is a valuable direction. **This will be a key part of our future work**, with the unwavering condition that any new data integration must strictly uphold the principle of real-world fidelity.
>
> This is a highly valuable question, as it directly probes the ultimate purpose behind our benchmark's design. As this point was raised by several reviewers, we have provided a more detailed and consolidated explanation in the ***public comment*** to all reviewers.
>
> **2. Response to weakness 2**
>
> To ensure we address your concern accurately, we are not entirely certain what specific types of "non-live alternative data sources" you have in mind. Based on our current understanding, one interpretation is the **inclusion of more static or "offline" datasets**. We would like to highlight that our benchmark does indeed incorporate such a scenario through the BEAR dataset (Lines 694-701, Appendix A.1). BEAR consists of a complete, high-frequency dataset collected from 80 rooms over a finite period. Our experiments on this dataset provide a direct evaluation on a high-quality offline source.
>
> We certainly agree that expanding the benchmark to include more high-fidelity offline datasets is a valuable direction for future research, and our extensible framework is explicitly designed to support such additions.
>
> However, our primary focus on live API streams for the majority of the datasets was a deliberate design choice to tackle the most critical and pervasive issues in existing benchmarks. As you astutely pointed out in the strengths, the "use of live API streams prevents contamination and ensures temporal realism." This principle is the cornerstone of our effort to build a truly leak-free and forward-looking evaluation standard. We believe that establishing this robust foundation was the most pressing and foundational contribution needed by the community at this time, providing a clear paradigm for how future datasets, both live and offline, should be curated.

---

> ### Author Response · Authors · 2025-11-21
> **Response to Reviewer xbyV (Part 2)**
>
> **3. Response to weakness 3**
>
> This is an key point that touches upon the essential characteristics a robust benchmark must possess to truly test the limits of modern forecasting models. Fidel-TS was explicitly designed to cover these diverse and challenging scenarios.
>
> First, regarding long-horizon forecasting, this is a central aspect of our experimental design. For all hourly datasets, our evaluation framework uses a substantial historical context (360 hourly steps, or half a month) to predict far into the future, with prediction lengths of {24, 168, 336, 720} steps, corresponding to {1 day, 1 week, 2 weeks, and 1 month}, respectively (Lines 327-329). **This setup ensures that our evaluation is firmly rooted in the long-term forecasting paradigm.**
>
> We interpret the rest of your comment as a question about **whether our benchmark's scope is diverse enough to cover scenarios with varying difficulty and temporal dynamics**. **The answer is yes**, and we'd like to highlight two key dimensions:
> * **(a) Simple vs. Critical/Difficult Data Domains**: The datasets within Fidel-TS intentionally span a range of complexities. As visualized in Appendix A.2 (Figure 2), the CCP and GREG datasets exhibit strong, clear periodic patterns. These represent relatively "simpler," more predictable domains. In sharp contrast, the NYTS (NYC Traffic Speed) dataset is characterized by significant noise and frequent, realistic periods of sensor downtime. This presents a much more challenging or "critical" forecasting task. This distinction is quantitatively confirmed by our results. As shown in Table 2 (page 6), under identical model configurations, the Mean Squared Error (MSE) for the NYTS dataset is consistently an order of magnitude higher than for CCP or GREG, validating its greater difficulty.
> * **(b) Adapting to Sudden Distributional Shifts**: Our use of real-world data streams ensures that such shifts are naturally present in the benchmark. These shifts are often driven by exogenous events, which is precisely where multimodal information becomes critical. For example, a sudden severe weather event (captured in the textual data) can cause a sharp, non-cyclical drop in traffic speed (a distributional shift in the time series). Our experimental results demonstrate the benchmark's ability to test for this. On the NYTS dataset, where traffic is highly susceptible to such shifts, causally-aware multimodal models show a distinct performance advantage over their unimodal counterparts (as seen in the comparison between Table 4 and Table 5). This performance lift indicates that the model is successfully leveraging the textual context to adapt to the distributional shift, a capability that our benchmark is uniquely positioned to evaluate.
>
> These analyses are valuable but are understated in our article. We commit to supplementing this part of the analysis in the Appendix.

---

> ### Author Response · Authors · 2025-11-26
> **Paper revision**
>
> Dear Reviewer,
>
> We have incorporated all the modifications committed to in our earlier response, and these changes are marked in blue in the revised manuscript. Regarding the specific questions you raised, we have summarized the details below for your convenience:
>
> 1. We updated the analysis of the data sample examples in Appendix A.2 (lines 837-850), with a focus on forecasting difficulty and distribution shifts in our data.
> 2. We provided a clearer explanation of why we deliberately chose weather in Section 3.1 (lines 140-150)
> 3. We updated the organization of the Experiment section, added a new Discussion section (Section 5), and fully presented the ablation study in Section 5.2 (lines 490-506). The results of ablation study demonstrate that for the BEAR dataset, more causally sound control events outperform weather conditions. This leads to our limitations and future works In Section 5.3 (lines 507-522).
>
> Thank you very much for your valuable comments and suggestions.

---

### Official Review · Reviewer_RYJU · 2025-11-03

**Soundness:** 3
**Presentation:** 3
**Contribution:** 3
**Rating:** 4
**Confidence:** 4

**Summary:**

The paper aims to overcome benchmarking limitations in the field by proposing a high-quality benchmark called Fidel-TS. It is a large scale benchmark containing 6 datasets (500k - 67million points) with high-frequency, continuously updated data, that is authentication protected. This gets around the pretraining data leakage problem -- where one is unsure whether an LLM was pretrained on the evaluation data. The benchmark also allows for forecasting over long prediction horizon and test sets to evaluate generalization.

**Strengths:**

- Continuously updated data streams from authentication protected APIs is a solid approach to protect against data leakage into LLM pretraining data. This is a unique aspect of this work

- One of the few benchmark papers in the field that protects against direct dataset memorization while still providing large scale benchmarking/evaluation (500k - 67 million points depending on the domain, table 1)

- The framework is designed to be extensible to other domains (asuch as ecnomic indicators or social media trends, as the authors state) and enables other researchers to add to this benchmark, which would benefit the community.

- Allows testing for generalization to new subjects.

**Weaknesses:**

- Authors mention that the primary source of exogenous data is weather data (scheduled maintenance is another). This limits the generality of the benchmark and is a bit too specific.

- In real world scenarios, many events occur that might affect forecasting. This information may be ingested by the model via searching news articles or other media, that might sometimes be informative and other times not. In extreme cases, the background/context might even be conflicting. How can we benchmark models in such scenarios. These questions are not addressed in the work and seem to be out of scope of the paper. In my opinion, this is a limitation of the work.

**Questions:**

- Can the authors explain the text on page 5: "Second, to address the unique computational challenges
of evaluating LLMs, we curate smaller subsets (denoted by the -mini suffix) constructed via
importance sampling. By selecting samples from the full test set where unimodal models outperform
multimodal ones and vice versa, we ensure both computational efficiency and fairness." The last sentence is unclear. Also, more details on the importance sampling would help.

- The pass rate is reported in tables but the text does not define what it means for a model to "pass"

---

> ### Author Response · Authors · 2025-11-21
> **Response to Reviewer RYJU (Part 1)**
>
> Thank you for your valuable feedback on our manuscript. We are grateful for the comments that Fidel-TS is "one of the few benchmark papers in the field that protects against direct dataset memorization while still providing large scale benchmarking/evaluation."
>
> **1. Response to the weaknesses**
>
> Our primary goal is not to create a benchmark with the broadest possible generalizability by simply incorporating more or increasingly complex text data. On the contrary, we observe that existing benchmarks, such as TimeMMD (the first multimodal time-series forecasting benchmark), do precisely this by using complex, multi-source text data. However, as we argue in our paper (Section 2.2, lines 94-120), this approach can lead to a significant data leakage problem, which misaligns with practical, real-world application scenarios. We believe that **proposing and adhering to the concept of real-world fidelity is a more critical and timely contribution than continuing the trend of unconstrained expansion of benchmark complexity**.
>
> Regarding the second weakness: We fully agree with your perspective. Our choice of weather data was deliberate, as it represents a universal text source that is relevant across numerous domains while strictly maintaining real-world fidelity. However, we concur that for any specific application domain, performance could be further enhanced by incorporating more domain-specific text, provided that the principle of real-world fidelity is upheld. **We have also mentioned this important direction for future research** in our "Limitations and Future Work" section (lines 477-485).
>
> These two questions are insightful that directly addresses the central core motivation and contribution of our paper. As this point was also raised by other reviewers, we have provided a more comprehensive and detailed discussion in the ***public comment***.

---

> ### Author Response · Authors · 2025-11-21
> **Response to Reviewer RYJU (Part 2)**
>
> In addition to the weaknesses, you also pointed out a few unclear concepts in our manuscript, specifically regarding **importance sampling** and the **LLM pass rate**. We sincerely apologize for this oversight and for not providing a clearer explanation in the initial submission. We would like to clarify these two points below.
>
> **2. Clarification on Importance Sampling**
>
> Our motivation for introducing importance sampling was to address the prohibitive inference cost of evaluating LLMs on our benchmark's test set, which contains a large number of high-frequency data points. Meanwhile, we need to ensure that the sampled test set is as similar in composition as possible to the full test set.
>
> Within the test set, samples vary in difficulty from a multimodal perspective. For some samples, the associated text is highly correlated with the time series, and leveraging this text can significantly improve prediction performance. For others, the text provides only marginal gains. A simple random sampling would fail to preserve this distribution of difficulty, leading to a potentially biased or unfair evaluation.
>
> To mitigate this, we devised an importance sampling strategy as follows:
> First, we selected two computationally less expensive models as proxies: a state-of-the-art unimodal model (PatchTST) and a multimodal model (FIATS). We evaluated both models on the entire test set. Based on their performance difference, we categorized the samples into two groups:
> * (a) samples where the multimodal model showed a significant performance gain over the unimodal (indicating high text relevance)
> * (b) samples where the improvement was marginal or negligible.
>
> After this classification, we sampled from each group using a fixed rate (e.g., for smaller dataset CCP, 10%; and for larger dataset NYTS, 1%) to construct the final -mini test set. This stratified approach ensures that the ratio of "text-sensitive" to "text-insensitive" samples in our -mini set faithfully reflects the distribution in the original full test set, thereby maintaining its integrity for evaluation.
>
> **3. Clarification on the LLM Pass Rate**
>
> The "pass rate" metric used in our work is conceptually analogous to the pass@k metric commonly employed in LLM reasoning and code-generation papers. In our specific setting, for each sample in the test set, we provide the LLM three attempts (i.e., k=3) to generate a prediction that strictly adheres to the required output format specified in our prompt. If the LLM produces a correctly formatted output within these three tries, the sample is considered a "pass"; If the LLM still fails to generate a format-compliant output after three attempts, it is marked as a "failure". **The final pass rate is therefore calculated as the percentage of total test samples for which the LLM successfully generated a correctly formatted output under this pass@3 condition.**
>
> As we observed in our process of experiment, The most common failure is that the LLM does not output predictions of the required length, often shorter than the output window (for example, it is required to output 24 timestamps but only outputs 23 steps in the end). Another common failure is that LLMs fails to comprehend the univariate/multivariate structure of the data, leading to an output matrix with an incorrect number of variables.
>
> We recognize that the details of "importance sampling" and "LLM pass rate" are essential for a complete understanding of our experiments. **We promise to amend the final version of our paper to include these explicit definitions**, ensuring that future readers do not face the same confusion.

---

> ### Author Response · Authors · 2025-11-26
> **Paper revision**
>
> Dear Reviewer,
>
> We have incorporated all the modifications committed to in our earlier response, and these changes are marked in blue in the revised manuscript. Regarding the specific questions you raised, we have summarized the details below for your convenience:
>
> 1. We supplemented the importance sampling algorithm to Appendix A.3 (lines 851-868)
> 2. We supplemented the explanation of llm pass rate to section 4.2 (line 305-309)
> 3. We provided a clearer explanation of why we deliberately chose weather in Section 3.1 (lines 140-150)
> 4. We updated the "limitation and future work" section (Section 5.3, lines 507-522), acknowledging the limitations of the weather and explaining the more causally sound text expansion and data filtering pipeline in the future under the real-world fidelity principle.
>
> Thank you very much for your valuable comments and suggestions.

---

### Author Response · Authors · 2025-11-21
**Our primary motivation and the choice of weather.**

We sincerely thank the reviewers for their thoughtful feedback and for recognizing the core strengths of our work. We find that a common question is whether it is possible to expand more data sources as text input for our benchmark, such as economy, news, etc. We believe this is a good opportunity to **clarify the primary motivation behind our design choices for Fidel-TS**.

As we demonstrated in our paper (Section 2, lines 72-121), the core intention of our work is to first address the foundational problems we have observed in existing benchmarks, which create a significant disconnect from real-world forecasting scenarios. These problems fall into two main categories:
* **Time Series Data Integrity**: As we mentioned in our paper (Section 2.1, lines 74-92), existing benchmarks are often outdated, small-scale, vague in structure and use low-frequency data. This hinders robust model evaluation.
* **Multimodal Data Fidelity**: This is a more subtle but equally critical issue. While the first multimodal benchmark commendably tried to include diverse text types like news and economic reports, this approach inadvertently introduced severe data quality issues. As we detail in our paper (Lines 041-043), these include pre-training contamination, temporal leakage (accessing information published after the event), and description leakage (text explicitly stating the ground-truth).

For example, a benchmark that retrospectively scrapes news articles analyzing a stock market dip to "predict" that same dip is creating a test scenario that is impossible in the real world. A model succeeding on such a task is not demonstrating forecasting ability but rather an ability to extract answers from post-hoc descriptions. Evaluating models in such unrealistic settings offers limited value for real-world applications.

**Establishing high fidelity is a necessary prerequisite for evaluating complex multimodal forecasting, not an alternative to it. Therefore, our immediate priority was to establish a benchmark with strict real-world fidelity**, rather than pursuing textual breadth at the cost of fidelity. This led us to select weather report as our primary textual modality. As a near-universal factor, weather reports are easily accessible in the real world, and weather forecasts are quite mature. Weather can affect changes in real-world systems such as power and traffic flow, but the changes in weather are independent of these systems and will not be affected by them. Also, the weather report can be known in advance and will not directly describe how the variables to be predicted will change below. This provides a general and representative task for evaluating multimodal models close to the real world.

We also recognized the need to move beyond a single source of textual data. Although common sense suggests that weather is closely linked to most real-world systems, the effectiveness of multimodal forecasting relies more on text that is highly relevant to the system. Such text tends to reflect system changes more accurately than weather-related information. We took effort to incorporated internal control events in the BEAR dataset. These events share the same crucial properties as weather forecasts, showcasing a different but equally valid type of multimodal input.

We wholeheartedly agree with the assessment that incorporating broader and more complex data sources is the crucial next step. This is precisely the path forward we envision and have explicitly outlined in our **Limitations and Future Work** section (Lines 478-485). As we state in our paper, this will require "novel methodologies that can filter and align these noisier data streams while upholding the principles of high fidelity" (Lines 482-484). Our extensible framework is designed to facilitate exactly this line of future research.

We hope this response clarifies that our deliberate focus on textual data quality is a necessary foundational step towards a richer, more complex benchmarking scenarios while following the principle of real-world fidelity.

---

### Author Response · Authors · 2025-12-02
**Summary of our rebuttal (Part 1)**

We thank the Area Chair and reviewers for their time and constructive feedback.

In our work, we propose Fidel-TS, a benchmark designed to address critical limitations in current unimodal and multimodal time series forecasting benchmarks, including their outdated nature, small scale, data leakage issues, and ambiguous variable structures. Our key contribution is the introduction of "high-fidelity benchmarking" principles for the community, aiming to create evaluation environments that more closely reflect real-world scenarios for models.

We are encouraged by some of the good feedbacks. **Regarding motivation**, reviewers praised the work as "well-motivated" (xbyV), validating our focus on real-world fidelity to address the leakage and simplicity issues in existing benchmarks. **Regarding methodology**, our principles (Integrity, Soundness, Clarity) were endorsed as a "solid approach" (RYJU) and "rigorous foundation" (Jj4v) that effectively "prevents contamination" (xbyV). **Regarding experiments**, reviewers affirmed that our strategy successfully "reveals hidden biases in prior evaluations" (Jj4v). **Regarding community impact**, reviewers highlighted that Fidel-TS fills a "crucial gap" (Jj4v) by providing a large-scale, leak-proof benchmark (RYJU) with a valued extensible framework (xbyV).

In addition to advantages, we actively address the issues raised by the reviewers:

1. **Justification for Weather Data (A Common Concern)**:
Reviewers questioned our decision to prioritize weather data over more complex texts like news. In our Public Comment ***"Our primary motivation and the choice of weather"***, we provided a detailed clarification. We emphasize that the priority of proposing the real-world fidelity principle is higher than that of expanding more textual data. Weather is a suitable choice since it avoids data leakage and is causally sound. We acknowledge its limitations as it may not be the most important factor driving system changes. This points to future work: expanding the use of more relevant textual data in specific domain while adhering to the principle of real-world fidelity.
2. **Implementation Details (Reviewer RYJU)**:
Reviewer RYJU pointed out missing definitions regarding "Importance Sampling" and "LLM Pass Rate." We have added these missing details to the revised manuscript (Appendix A.3 and Section 5.1). **Although Reviewer RYJU gave a rating of 4, his/her concerns were primarily factual questions regarding these missing definitions. We believe that by completing these details, we have fully resolved his/her doubts.**
3. **Offline Data and Deep Analysis (Reviewer xbyV)**:
To address Reviewer xbyV’s inquiry about non-live data, we clarified the role of the BEAR dataset as our high-quality, completely offline benchmark component. Furthermore, we adopted the reviewer's suggestion for deeper data characterization, updating Appendix A.2 to clearly visualize distribution shifts and data difficulty across our datasets.
4. **Reproducibility, Causality, and Computational Cost Analyses (Reviewer Jj4v)**:
We addressed reproducibility concerns by highlighting our use of stable official APIs and providing static snapshots. Regarding causal soundness, we clarified that since formal theoretical proof is intractable for complex systems, so we rely on robust common-sense priors. Finally, while we adopted the suggestion to add computational cost analyses (Section 5.1), we respectfully noted that model-specific ablations fall outside the scope of our benchmark-driven study. **After we revised the article, reviewer Jj4v indicated that our response addressed his/her questions.**
5. **Additional Experiments and Expressions of Key Words (Reviewer 1Gia)**:
We enhanced the depth of our evaluation by adding LLM reasoning examples and computational cost comparisons (same as Reviewer Jj4v). Regarding other prompting strategies (LLMP, LLMTime), we clarified that they fall outside our benchmark-driven scope and listed in our limitation and future work. Additionally, we refined our presentation to "Leak-free and Causally Sound Design", "Temporal Leakage" and strictly aligning the main text with the pipeline figure to prevent misunderstanding.

---

> ### Author Response · Authors · 2025-12-02
> **Summary of our rebuttal (Part 2)**
>
> During our rebuttal, we were genuinely concerned about the review quality from Reviewer 1Gia. Many of the comments rely on basic factual mistakes and subjective opinions rather than technical issues.
>
> The most striking example is the accusation that our citation of “FIATS” is a repeated misspelling of “FITS.” These are two different models that are correctly cited in our paper. Calling this “unprofessional” without even checking the references shows a lack of basic due diligence.
>
> Beyond these factual errors, the review also reveals a misunderstanding of standard time series tasks. The reviewer incorrectly describes the classic Long-Term Time Series Forecasting task as univariate, confuses variable structure with multimodal forecasting, and dismisses zero-shot evaluation protocols that are central to modern Time Series Foundation Models and LLMs. Furthermore, The reviewer wrongly believed that whether a model could be evaluated depended on the type of benchmark rather than the task. These points directly contradict well-established practices in the field.
>
> The review further relies on subjective and ungrounded criticism, such as calling our writing “extremely poor” without giving a single example. It also overlooks the main motivation of our work. As a result, The reviewer misunderstood the rationality of our weather selection and questioned that our ablation study did not demonstrate the causal soundness principle but rather undermined the quality of the benchmark.

---

> ### Author Response · Authors · 2025-12-02
> **Summary of our rebuttal (Part 3)**
>
> We appreciate all constructive and insightful questions from the whole community. Our method is meant to be an initial, exploratory step: a first attempt to highlight limitations in current benchmarks and propose a methodology for high-fidelity benchmarking. However, we are concerned that the comments from Reviewer 1Gia were not constructive. Several points were based on factual misunderstandings, reflected a lack of familiarity with the domain, and included subjective or even hostile remarks that fall outside the norms of professional scientific discussion. We are deeply concerned about this situation.

---

### Meta-Review · Area_Chair_Wn94 · 2026-01-06

**Summary:**

The paper proposes Fidel-TS, a high-fidelity benchmark for multimodal time series forecasting designed to address issues like data leakage and pre-training contamination in existing datasets. Reviewers generally appreciated the motivation behind creating a leak-free, causally sound benchmark using live API data. However, there are concerns regarding the scope and implementation of the benchmark. Specifically, reviewers questioned the heavy reliance on weather data as the primary text modality, arguing it limits the benchmark's generalizability compared to more complex text sources like news or economic reports. Additionally, there were strong reservations about the evaluation methodology, including the appropriateness of testing LLMs zero-shot on this specific type of data and the lack of comprehensive comparisons with diverse prompting strategies.

**Reviewer Concerns:**

The rebuttal addressed some factual misunderstandings, such as the distinction between the FITS and FIATS models (answering Reviewer 1Gia) and provided missing definitions for "importance sampling" and "LLM pass rate" (answering Reviewer RYJU). The authors also clarified their rationale for prioritizing weather data to ensure causal soundness. However, the core criticism regarding the limited diversity of text modalities remains a major outstanding concern. Reviewers 1Gia and xbyV were not fully convinced that restricting the benchmark primarily to weather data is sufficient for a general-purpose multimodal benchmark, even if it ensures "fidelity." Furthermore, the debate over the fairness of comparing zero-shot LLMs against trained multimodal models and the lack of evaluation with advanced prompting techniques (like LLMTime) was not fully resolved to the satisfaction of the critical reviewer.

**Reviewer Scores:**

Reviewer Jj4v explicitly stated that their concerns were addressed, suggesting they would likely maintain or slightly improve their positive score (6). Reviewer RYJU had factual questions resolved but expressed fundamental concerns about the benchmark's generality, likely keeping their score at 4. Reviewer xbyV, who rated the paper a 6, might maintain this score given their positive view of the motivation, but the unaddressed weakness regarding domain diversity could prevent a higher rating. Reviewer 1Gia, who gave a strong reject (2), is unlikely to change their score significantly, as they viewed the experimental design and presentation as fundamentally flawed, despite the authors' clarifications on factual errors.

---

### Decision · Program_Chairs · 2026-01-26

Reject